**Investigation**

# Interpreting SNP heritability in admixed populations

Jinguo Huang [ID] ,[1,2,†] Nicole Kleman [ID] ,[3,†] Saonli Basu [ID] ,[4] Mark D. Shriver [ID] ,[2] Arslan A. Zaidi [ID] [3,5,*]

[1]Bioinformatics and Genomics, Huck Institutes of the Life Sciences, Pennsylvania State University, University Park, PA 16802, USA
[2]Department of Anthropology, Pennsylvania State University, University Park, PA 16802, USA
[3]Department of Genetics, Cell Biology, and Development, University of Minnesota, 6-160 Jackson Hall, 321 Church St. SE, Minneapolis, MN 55455, USA
[4]Department of Biostatistics, University of Minnesota, Minneapolis, MN 55455, USA
[5]Institute of Health Informatics, University of Minnesota, Minneapolis, MN 55455, USA

*Corresponding author: Department of Genetics, Cell Biology, and Development, University of Minnesota, 6-160 Jackson Hall, 321 Church St. SE, Minneapolis, MN 55455, USA. Email: aazaidi@umn.edu
[†]These authors contributed equally to this work.

Single-nucleotide polymorphism (SNP) heritability ($h^2_{snp}$) is defined as the proportion of phenotypic variance explained by genotyped SNPs and is believed to be a lower bound of heritability ($h^2$), being equal to it if all causal variants are genotyped. Despite the simple intuition behind $h^2_{snp}$, its interpretation and equivalence to $h^2$ is unclear, particularly in the presence of admixture and assortative mating. Here, we use analytical theory and simulations to describe the behavior of $h^2$ and three widely used random-effect estimators of $h^2_{snp}$—genome-wide restricted maximum likelihood, Haseman–Elston regression, and LD score regression—in admixed populations. We show that $h^2_{snp}$ estimates can be biased in admixed populations, even if all causal variants are genotyped and in the absence of confounding due to shared environment. This is largely because admixture generates directional LD, which contributes to the genetic variance, and therefore to heritability. Random-effect estimators of $h^2_{snp}$, because they assume that SNP effects are independent, do not capture the contribution, which can be positive or negative depending on the genetic architecture, leading to under- or over-estimates of $h^2_{snp}$ relative to $h^2$. For the same reason, estimates of local ancestry heritability ($\hat{h}^2_\gamma$) are also biased in the presence of directional LD. We describe this bias in $\hat{h}^2_{snp}$ and $\hat{h}^2_\gamma$ as a function of admixture history and the genetic architecture of the trait, clarifying their interpretation and implication for genome-wide association studies and polygenic prediction in admixed populations.

Keywords: heritability; admixture; complex traits; population structure; statistical genetics; GREML; HE regression; LDSC

## Introduction

The ability to estimate (narrow-sense) heritability ($h^2$) from unrelated individuals was a major advance in genetics. Traditionally, $h^2$ was estimated from family-based studies in which the phenotypic resemblance between relatives could be modeled as a function of their expected genetic relatedness (Lynch and Walsh 1998). However, this approach was limited to analysis of closely related individuals where pedigree information is available and the realized genetic relatedness is not too different from expectation (Visscher et al. 2008). With the advent of genome-wide association studies (GWAS), we hoped that many of the variants underlying this heritability would be uncovered. However, when genome-wide significant single-nucleotide polymorphisms (SNPs) explained a much smaller fraction of the phenotypic variance, it became important to explain the missing heritability—were family-based estimates inflated or were GWAS just underpowered, limited by variant discovery?

Yang et al. (2010) made the key insight that one could estimate the portion of $h^2$ tagged by genotyped SNPs, regardless of whether or not they were genome-wide significant, by exploiting the subtle variation in the realized genetic relatedness among apparently unrelated individuals (Yang et al. 2010; Yang, Manolio, et al. 2011; Yang, Lee, et al. 2011). This quantity came to be known colloquially as "SNP heritability" ($h^2_{snp}$), and it is believed to be equal

to $h^2$ if all causal variants are included among genotyped SNPs (Yang et al. 2010). Indeed, estimates of $h^2_{snp}$ explain a much larger fraction of trait heritability than GWAS SNPs (Yang et al. 2010), approaching family-based estimates of $h^2$ when whole-genome sequence data, which captures rare variants, are used (Wainschtein et al. 2022). This has made it clear that GWAS have yet to uncover more variants with increasing sample size. Now, $h^2_{snp}$ has become an important aspect of the design of genetic studies and is often used to define the power of variant discovery in GWAS and the upper limit of polygenic prediction accuracy.

Despite the utility and simple intuition of $h^2_{snp}$, there is much confusion about its interpretation and equivalence to $h^2$, particularly in the presence of population structure and assortative mating (Browning SR and Browning 2011; Goddard et al. 2011; Kumar et al. 2016a; Yang et al. 2016; Border et al. 2022; Lin Z et al. 2022). But much of the discussion of heritability in structured populations has focused on biases in $\hat{h}^2_{snp}$—the estimator—due to confounding effects of shared environment and linkage disequilibrium (LD) with other variants (Visscher et al. 2010; Browning SR and Browning 2011; Kumar et al. 2016a; Yang et al. 2016; Lin Z et al. 2022). There is comparatively little discussion, at least in human genetics, on the fact that LD due to population structure also contributes to genetic variance, and therefore, is a component of heritability (Lynch and Walsh 1998; Rawlik et al. 2020;

Lara *et al.* 2021). We think this is at least partly due to the fact that most studies are carried out in cohorts with primarily European ancestry, where the degree of population structure is minimal and large effects of LD can be ignored. However, that is not the case for diverse, multi-ethnic cohorts, which have historically been underrepresented in genetic studies, but thanks to a concerted effort in the field, are now becoming increasingly common (The All of Us Research Program Investigators 2019; Wojcik *et al.* 2019; Ben-Eghan *et al.* 2020; Fatumo *et al.* 2022; Verma *et al.* 2022; Johnson *et al.* 2023; Sohail *et al.* 2023). The complex structure in these cohorts also brings unique methodological challenges, and it is imperative that we understand whether existing methods, which have largely been evaluated in more homogeneous groups, generalize to more diverse cohorts.

Our goal in this paper is to study the behavior of $h^2$ and $\hat{h}^2_{\text{snp}}$ in admixed populations. How should we interpret $\hat{h}^2_{\text{snp}}$ in the ideal situation where causal variants are directly genotyped? Is it an unbiased estimate of $h^2$? To answer these questions, we derived a general expression for the genetic variance in admixed populations, decomposing it in terms of the contribution of population structure, which influences both the genotypic variance at individual loci and the LD across loci. We used theory and simulations to show that $\hat{h}^2_{\text{snp}}$ estimated with genome-wide restricted maximum likelihood (GREML) (Yang *et al.* 2010; Yang, Lee, *et al.* 2011), Haseman–Elston (HE) regression (Haseman and Elston 1972), and linkage disequilibrium score regression (LDSC) (Bulik-Sullivan *et al.* 2015)—three widely used approaches—can be biased in admixed and other structured populations, even in the absence of confounding and when all causal variants are genotyped. We explain this in terms of the discrepancy between the model assumed in $\hat{h}^2_{\text{snp}}$ estimation and the generative model from which the genetic architecture of the trait in the population may have been sampled. We describe the bias in $\hat{h}^2_{\text{snp}}$ as a function of admixture history and genetic architecture and discuss its implications for GWAS and polygenic prediction accuracy.

## Model
### Genetic architecture
We begin by describing a generative model for the phenotype. Let $y = g + e$, where $y$ is the phenotypic value of an individual, $g$ is the genotypic value, and $e$ is random error. We assume additive effects such that $g = \sum_{i=1}^{m} \beta_i x_i$ where $\beta_i$ is the effect size of the ith biallelic locus (out of a total $m$ causal loci) and $x_i \in \{0, 1, 2\}$ is the number of copies of the trait-increasing allele. Importantly, the effect sizes are fixed quantities and differences in genetic values among individuals are due to random variation in genotypes. Note, that this is different from the random-effects model assumed by many heritability estimators where genotypes are fixed and effect sizes are random (de los Campos *et al.* 2015).

We denote the mean, variance, and covariance with $\mathbb{E}(.)$, $\mathbb{V}(.)$, and $\mathbb{C}(.)$, respectively, where the expectation is measured over random draws from the population rather than random realizations of the evolutionary process. We can express the additive genetic variance of a quantitative trait as follows:

$$V_g = \mathbb{V}\left(\sum_{i=1}^{m} \beta_i x_i\right) = \sum_{i=1}^{m} \beta_i^2 \mathbb{V}(x_i) + \sum_{j \neq i} \beta_i \beta_j \mathbb{C}(x_i, x_j).$$

Here, the first term represents the contribution of individual loci (genic variance), and the second term is the contribution of linkage disequilibrium (LD contribution). We make the assumption that

loci are unlinked and, therefore, the LD contribution is entirely due to population structure. We describe the behavior of $V_g$ in a population that is a mixture of two previously isolated populations A and B that diverged from a common ancestor. To do this, we denote $\theta$ as the fraction of the genome of an individual with ancestry from population A. Thus, $\theta = 1$ if the individual is from population A, 0 if they are from population B, and $\theta \in (0, 1)$ if they are admixed. Then, $V_g$ can be expressed in terms of ancestry as (Appendix):

$$V_g = 2\mathbb{E}(\theta) \sum_{i=1}^{m} \beta_i^2 f_i^A (1 - f_i^A) + 2\{1 - \mathbb{E}(\theta)\} \sum_{i=1}^{m} \beta_i^2 f_i^B (1 - f_i^B) \quad (1.1)$$

$$+ 2\mathbb{E}(\theta)\{1 - \mathbb{E}(\theta)\} \sum_{i=1}^{m} \beta_i^2 (f_i^A - f_i^B)^2 \quad (1.2)$$

$$+ 2\mathbb{V}(\theta) \sum_{i=1}^{m} \beta_i^2 (f_i^A - f_i^B)^2 \quad (1.3)$$

$$+ 4\mathbb{V}(\theta) \sum_{i \neq j} \beta_i \beta_j (f_i^A - f_i^B)(f_j^A - f_j^B), \quad (1.4)$$

where $f_i^A$ and $f_i^B$ are the allele frequencies in populations A and B, respectively, and $\mathbb{E}(\theta)$ and $\mathbb{V}(\theta)$ are the mean and variance of individual ancestry, respectively. The sum of the first three terms represents the genic variance and the last term represents the LD contribution.

### Demographic history
From Equation 1, it is clear that, conditional on the genetic architecture in the source populations ($\beta, f^A, f^B$), $V_g$ is a function of the mean, $\mathbb{E}(\theta)$, and variance, $\mathbb{V}(\theta)$, of individual ancestry in the admixed population. We consider two demographic models that affect $\mathbb{E}(\theta)$ and $\mathbb{V}(\theta)$ in qualitatively different ways. In the first model, the source populations meet once $t$ generations ago (we refer to this as $t = 0$) in proportions $p$ and $1 - p$, after which there is no subsequent admixture (Fig. 1a). In the second model, there is continued gene flow in every generation from one of the source populations such that the mean overall amount of ancestry from population A is the same as in the first model (Fig. 1a). For brevity, we refer to these as the hybrid isolation (HI) and continuous gene flow (CGF) models, respectively, following Pfaff *et al.* (2001). $\mathbb{V}(\theta)$ is also affected by ancestry-based assortative mating, where individuals are more likely to partner with others of similar ancestry. We refer to this simply as assortative mating for brevity and model this following Zaitlen *et al.* (2017) using a parameter $P \in (0, 1)$, which represents the correlation of the ancestry of individuals across mating pairs in the population.

Under these conditions, the behavior of $\mathbb{E}(\theta)$ and $\mathbb{V}(\theta)$ has been described previously (Verdu and Rosenberg 2011; Zaitlen *et al.* 2017) (Fig. 1b and c). Briefly, in the HI model, $\mathbb{E}(\theta)$ remains constant at $p$ in the generations after admixture as there is no subsequent gene flow. $\mathbb{V}(\theta)$ is at its maximum at $t = 0$ when each individuals carries chromosomes either from population A or B, but not both. This genome-wide correlation in ancestry breaks down in subsequent generations as a function of mating, independent assortment, and recombination, leading to a decay in $\mathbb{V}(\theta)$, the rate depending on the strength of assortative mating (Fig. 1c). In the CGF model, both $\mathbb{E}(\theta)$ and $\mathbb{V}(\theta)$ increase with time as new chromosomes are introduced from the source populations. But while $\mathbb{E}(\theta)$ continues to increase monotonically, $\mathbb{V}(\theta)$ will plateau and decrease due to the countervailing effects of independent assortment and recombination which redistribute ancestry in the population, approaching zero at equilibrium if there is no more gene flow and the population is mating randomly. $\mathbb{V}(\theta)$ provides

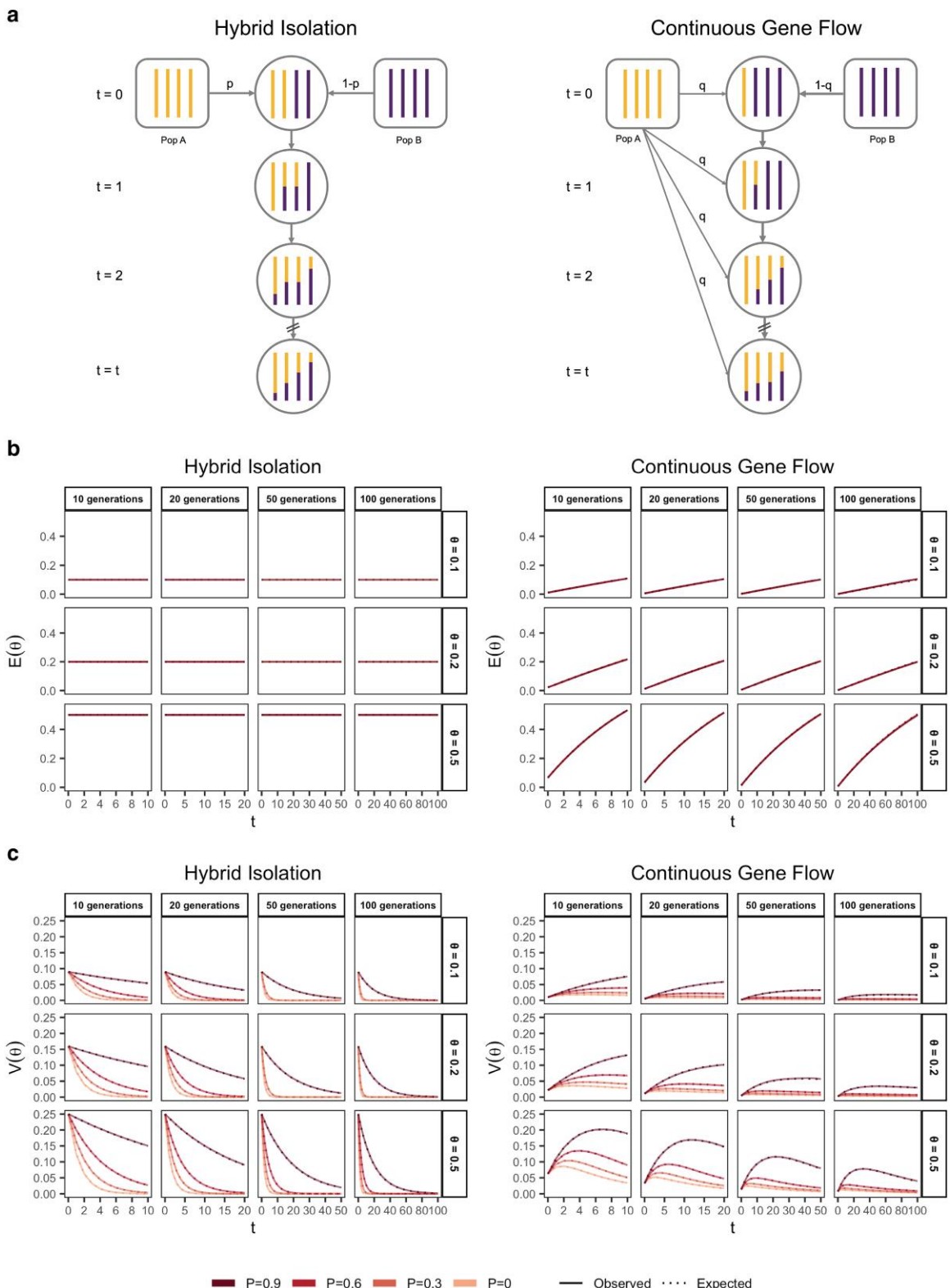

**Fig. 1.** The behavior of mean and variance of individual ancestry as a function of admixture history. a) Shows the demographic models under which simulations were carried out. Admixture might occur once (hybrid isolation, HI, left column) or continuously (continuous gene flow, CGF, right column). b) The mean individual ancestry, $\mathbb{E}(\theta)$ remains constant over time in the HI model and increases in the CGF model with continued gene flow. c) The variance in individual ancestry, $\mathbb{V}(\theta)$ is maximum at $t = 0$ in the HI model, decaying subsequently. $\mathbb{V}(\theta)$ increases with gene flow in the CGF model and will subsequently decrease with time. P measures the strength of assortative mating, which slows the decay of $\mathbb{V}(\theta)$. P = 0.6 is missing for simulations run for 50 and 100 generations and $\theta \in \{0.1, 0.2\}$ due to the difficulty in finding mate pairs (Methods).

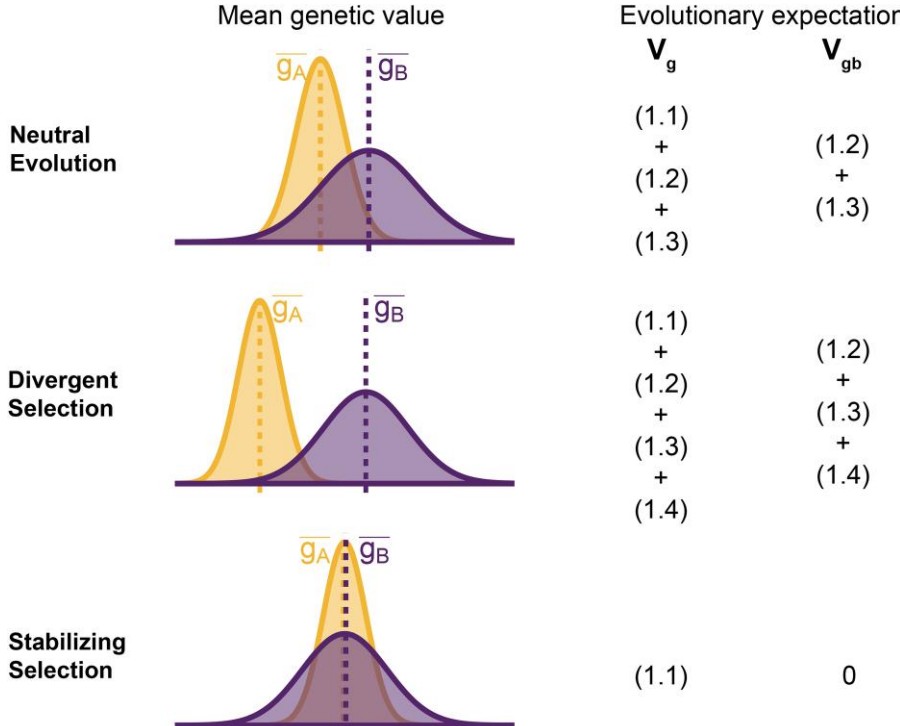

**Fig. 2.** Decomposing genetic variance in a two-population system. The plot illustrates the expected distribution of genetic values in two populations under different selective pressures and the terms on the right list the total ($V_g$) and between-population genetic variance ($V_{gb}$) expected over the evolutionary process. For neutrally evolving traits (top row), we expect there to be an absolute difference in the mean genetic values ($|\bar{g}_A - \bar{g}_B|$) that is proportional to $F_{ST}$. For traits under divergent selection (middle), $|\bar{g}_A - \bar{g}_B|$ is expected to be greater than that expected under genetic drift. For traits under stabilizing selection, $|\bar{g}_A - \bar{g}_B|$ will be less than that expected under genetic drift, and zero in the extreme case.

an intuitive and quantitative measure of the degree of population structure (along the axis of ancestry) in admixed populations.

## Results
### Genetic variance in admixed populations

To understand the expectation of genetic variance in admixed populations, it is first worth discussing its behavior in the source populations. In Equation 1, the first term represents the within-population component ($V_{gw}$) and the last three terms altogether represent the component of genetic variance between populations A and B ($V_{gb}$). Note that $V_{gb} = \frac{(\bar{g}_A - \bar{g}_B)^2}{2}$ is positive only if there is a difference in the mean genotypic values (Fig. 2). This variance increases with increasing $F_{ST}$, i.e. genetic divergence, between the two populations. While $\beta_i^2 (f_i^A - f_i^B)^2$ is expected to increase monotonically with $F_{ST}$, $\beta_i \beta_j (f_i^A - f_i^B)(f_j^A - f_j^B)$ is expected to be zero under neutrality because the direction of frequency change will be uncorrelated across loci. In this case, the LD contribution, i.e. (1.4), is expected to be zero and $V_{gb} = (1.1) + (1.2) + (1.3)$ 1.11.21.3. However, this is true only in expectation over the evolutionary process and the realized LD contribution may be nonzero even for neutral traits.

For traits under selection, the LD contribution is expected to be greater or less than zero, depending on the type of selection. Under divergent selection, trait-increasing alleles will be systematically more frequent in one population over the other, inducing positive LD across loci (Corre and Kremer 2003; Berg and Coop 2014), increasing the LD contribution, i.e. term (1.4). Stabilizing selection, on the other hand, induces negative LD (Bulmer 1971; Yair and Coop 2022). In the extreme case, the mean genetic

values of the two populations are exactly equal and $V_{gb} = (1.2) + (1.3) + (1.4) = 0$ 1.21.31.4. For this to be true, (1.4) has to be negative and equal to (1.2) + (1.3), which are both positive, and the total genetic variance is reduced to the within-population variance, i.e. term (1.1) (Fig. 2). This is relevant because, as we show in the following sections, the behavior of the genetic variance in admixed populations depends on the magnitude of $V_{gb}$ between the source populations.

We illustrate this by tracking the genetic variance in admixed populations for two traits, both with the same mean $F_{ST}$ at causal loci but with different LD contributions (term 1.4): one where the LD contribution is positive (trait 1) and the other where it is negative (trait 2). Thus, traits 1 and 2 can be thought of as examples of phenotypes under divergent and stabilizing selection, respectively, and we refer to them as such from hereon. To simulate the genetic variance of such traits, we drew the allele frequencies ($f^A$ and $f^B$) in populations A and B for 1,000 causal loci with $F_{ST} \approx 0.2$ using the Balding–Nichols model (Balding and Nichols 1995). We assigned effect sizes such that each locus contributes equally to the genetic variance, i.e. $\beta = \frac{1}{\sqrt{2m\bar{f}(1-\bar{f})}}$ where $\bar{f}$ is the mean allele frequency between the two populations, and $m$ is the number of loci. To simulate positive and negative LD, we permuted the effect signs across variants 100 times and selected the combinations that gave the most positive and negative LD contribution to represent the genetic architecture of traits that might be under directional (trait 1) and stabilizing (trait 2) selection, respectively (Methods). We simulated the genotypes of 10,000 individuals under the HI and CGF models for $t \in \{10, 20, 50, 100\}$ generations postadmixture and calculated genetic values for both traits using $g = \sum_{i=1}^{m} \beta_i x_i$, where $m = 1,000$ (Method). The observed genetic

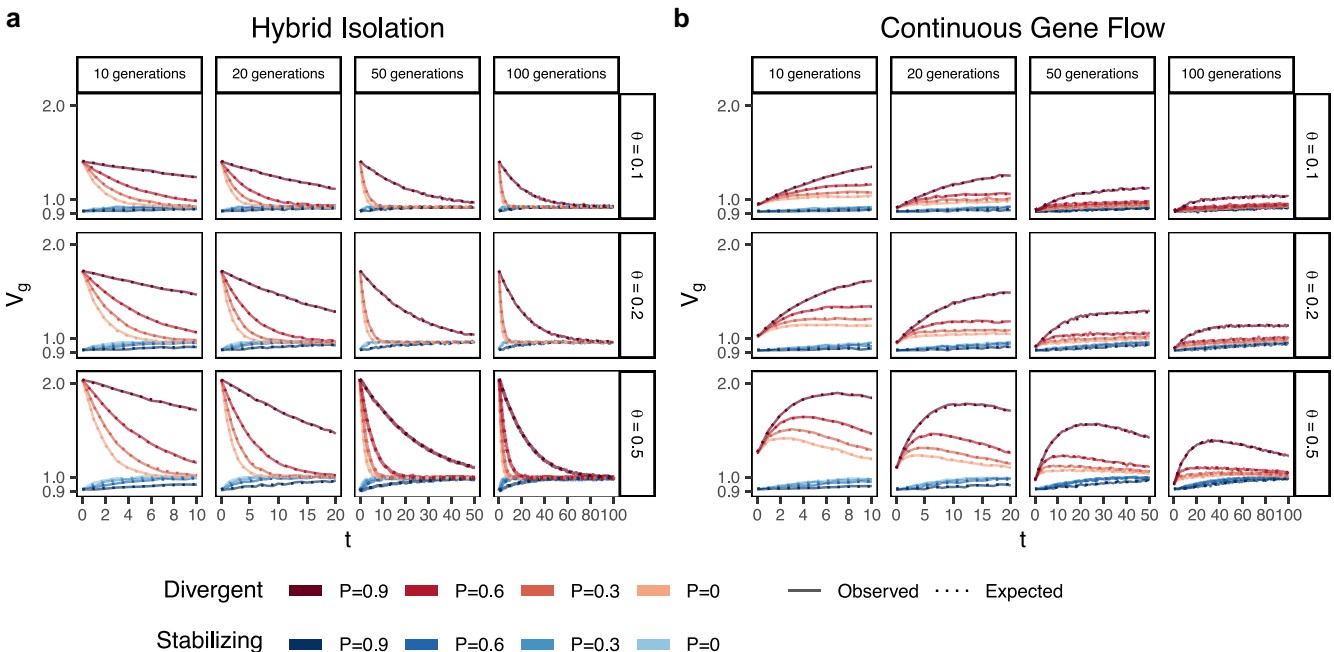

**Fig. 3.** Genetic variance in admixed populations under the a) HI and b) CGF models. Dotted lines represent the expected genetic variance based on Equation (1) and solid lines represent results of simulations averaged over 10 replicates. P = 0.6 is missing for simulations run for 50 and 100 generations and $\theta \in \{0.1, 0.2\}$ due to the difficulty in finding mate pairs (Methods).

variance at any time can then be calculated simply as the variance in genetic values, i.e. $V_g = \mathbb{V}(g)$.

In the HI model, $\mathbb{E}(\theta)$ does not change (Fig. 1b) so terms (1.1) and (1.2) are constant through time. Terms (1.3) and (1.4) decay towards zero as the variance in ancestry goes to zero and $V_g$ ultimately converges to (1.1) + (1.2) (Fig. 3). This equilibrium value is equal to the $\mathbb{E}(V_g|\theta)$ (Appendix) and the rate of convergence depends on the strength of assortative mating, which slows the rate at which $\mathbb{V}(\theta)$ decays. $V_g$ approaches equilibrium from a higher value for traits under divergent selection and lower value for traits under stabilizing selection because of positive and negative LD contributions, respectively, at $t = 0$ (Fig. 3). In the CGF model, $V_g$ increases initially for both traits with increasing gene flow (Fig. 3). This might seem counter-intuitive at first because gene flow increases admixture LD, which leads to more negative values of the LD contribution for traits under stabilizing selection (Supplementary Fig. 1). But this is outweighed by positive contributions from the genic variance—terms (1.1) + (1.2) + (1.3)—all of which initially increase with gene flow (Supplementary Fig. 1). After a certain point, the increase in $V_g$ slows down as any increase in $\mathbb{V}(\theta)$ due to gene flow is counterbalanced by recombination and independent assortment. Ultimately, $V_g$ will decrease if there is no more gene flow, reaching the same equilibrium value as in the HI model, i.e. $\mathbb{E}(V_g|\theta) = (1.1) + (1.2)1.11.2$. Because the loci are unlinked, we refer to the sum (1.3) + (1.4) as the contribution of population structure.

## GREML estimation

In their original paper, Yang *et al.* (2010) defined $h^2_{\mathrm{snp}}$ as the variance explained by genotyped SNPs and not as heritability (Yang *et al.* 2010). This is because $h^2$ is the genetic variance explained by causal variants, which are unknown. Genotyped SNPs may not overlap with or tag all causal variants and thus, $h^2_{\mathrm{snp}}$ is understood to be a lower bound of $h^2$, both being equal if causal variants are directly genotyped (Yang *et al.* 2010). Our goal is to

demonstrate (1) that this may not be true in structured populations, (2) quantify the bias in $\hat{h}^2_{\mathrm{snp}}$, and (3) understand the source and behavior of this bias even in the ideal situation when causal variants are genotyped.

We first used GREML, implemented in GCTA (Yang *et al.* 2010; Yang, Lee, *et al.* 2011), to estimate the genetic variance for our simulated traits. GCTA assumes the following model: $\mathbf{y} = \mathbf{Zu} + \epsilon$ where $\mathbf{Z}$ is an $n \times m$ standardized genotype matrix such that the genotype of the $k$th individual at the $i$th locus is $z_{ik} = \frac{x_{ik} - 2f_i}{\sqrt{2f_i(1-f_i)}}$, $f_i$ being the allele frequency. The SNP effects corresponding to the scaled genotypes are assumed to be random and independent such that $\mathbf{u} \sim \mathcal{N}(0, \mathbf{I}\frac{\sigma^2_u}{m})$ and $\epsilon \sim \mathcal{N}(0, \mathbf{I}\sigma^2_\epsilon)$ is random environmental error. Then, the phenotypic variance can be decomposed as:

$$\mathbb{V}(\mathbf{y}) = \mathbb{V}(\mathbf{Zu}) + \mathbb{V}(e)$$
$$= \frac{\mathbf{ZZ}'}{m}\sigma^2_u + \mathbf{I}\sigma^2_\epsilon,$$

where $\frac{\mathbf{ZZ}'}{m}$ is the genetic relationship matrix (GRM), the variance components $\sigma^2_u$ and $\sigma^2_\epsilon$ are estimated using restricted maximum likelihood, and $\hat{h}^2_{\mathrm{snp}}$ is calculated as $\frac{\hat{\sigma}^2_u}{\hat{\sigma}^2_u + \hat{\sigma}^2_\epsilon}$ (Supplementary Fig. 2). We are interested in asking whether $\hat{\sigma}^2_u$ is an unbiased estimate of $V_g$. To answer this, we constructed the GRM with causal variants and estimated $\hat{\sigma}^2_u$ using GCTA (Yang *et al.* 2010; Yang, Manolio, *et al.* 2011).

GCTA under- and over-estimates the genetic variance in admixed populations for traits under divergent (trait 1) and stabilizing selection (trait 2), respectively, when there is population structure, i.e. when $\mathbb{V}(\theta) > 0$ (Fig. 4a). One reason for this bias is that the GREML model assumes that the effects are independent, and therefore the LD contribution is zero. This, as discussed in the previous section, is not true for traits under divergent or stabilizing selection between the source populations, and only true for neutral traits in expectation. Because of this, $\hat{\sigma}^2_u$ does not capture the LD contribution,

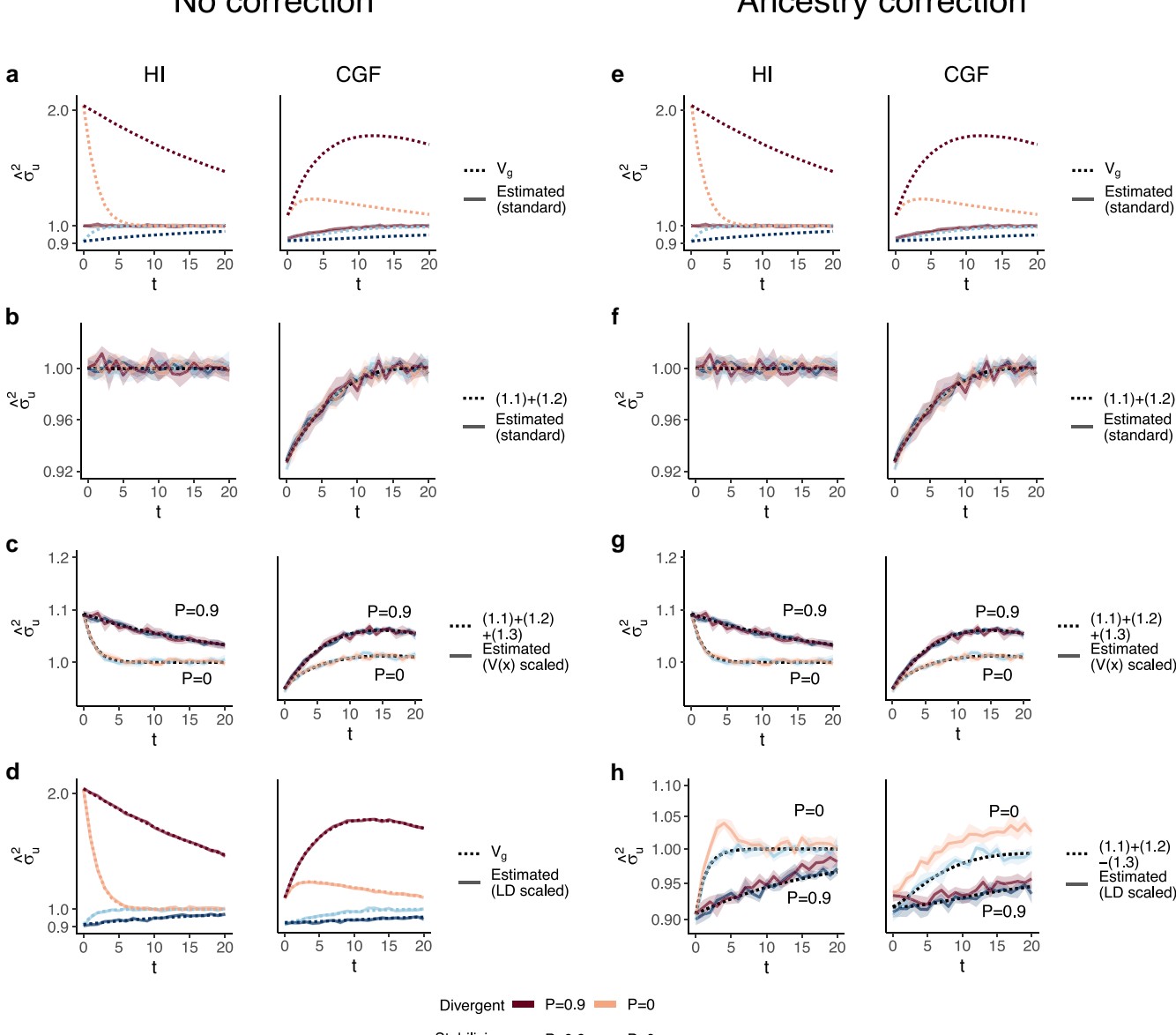

**Fig. 4.** The behavior of GREML estimates of the genetic variance ($\hat{\sigma}_u^2$) in admixed populations under the HI (left column) and CGF (right column) models either without (a–d) or with (e–h) individual ancestry as a fixed effect. The solid lines represent estimates from simulated data averaged across 10 replicates. P indicates the strength of assortative mating. The shaded area represents the 95% confidence bands generated by bootstrapping (sampling with replacement 100 times) the point estimate reported by GCTA. The dotted lines either represent the expected variance in the population based on Equation 1 (a and b) or the expected estimate for three different ways of scaling genotypes (b–d and f–h). (a–b and e–f) show the behavior of $\hat{\sigma}_u^2$ for the default scaling, (c, g) shows $\hat{\sigma}_u^2$ when the genotype at a locus is scaled by its sample variance ($\mathbb{V}(x)$ scaled), and (d, h) when it is scaled by the sample covariance (LD scaled).

i.e. term (1.4) (Fig. 4a). But $\hat{\sigma}_u^2$ can be biased even if the LD contribution is zero if the genotypes are scaled with $\sqrt{2f_i(1-f_i)}$—the standard practice—where $f_i$ is the frequency of the allele in the population. This scaling assumes that $\mathbb{V}(x_i) = 2f_i(1-f_i)$, which is true only if the population were mating randomly. In an admixed population $\mathbb{V}(x_i) = 2f_i(1-f_i) + 2\mathbb{V}(\theta)(f_i^A - f_i^B)^2$, where $f_i$, $f_i^A$, and $f_i^B$ correspond to frequency in the admixed population, and source populations, A and B, respectively (Appendix). Alternatively, if the genotypes are scaled, $\mathbb{V}(z_i) = 1 + 2\mathbb{V}(\theta)F_{st}^{(i)}$ where $F_{st}^{(i)}$ is the $F_{st}$ at the ith locus. We show that this assumption biases $\hat{\sigma}_u^2$ downwards by a factor of $2\mathbb{V}(\theta)(f_i^A - f_i^B)^2$ (or $2\mathbb{V}(\theta)F_{st}^{(i)}$ if genotypes are scaled)—term (1.3) (Fig. 4b, Appendix). Thus, with the standard scaling, $\hat{\sigma}_u^2$ gives a biased

estimate in the presence of population structure, even of the genic variance.

The overall bias in $\hat{\sigma}_u^2$ is determined by the relative magnitude and direction of terms (1.3) and (1.4), both of which are functions of $\mathbb{V}(\theta)$, and therefore, of the degree of structure in the population. The contribution of term (1.3) will be modest, even in highly structured populations (Supplementary Fig. 1) and therefore, the overall bias is largely driven by the LD contribution. If there is no more gene flow, $\mathbb{V}(\theta)$ will ultimately go to zero and $V_g$ will converge towards $\hat{\sigma}_u^2$. Thus, $\hat{\sigma}_u^2$ is more accurately interpreted as the genic variance or the genetic variance expected if the LD contribution were zero and if the population were mating randomly. In other words, $\mathbb{E}(\hat{\sigma}_u^2) = (1.1) + (1.2) \neq V_g 1.11.2$ (Fig. 4b).

In principle, we can recover the missing components of $V_g$ by scaling the genotypes appropriately. For example, we can recover term (1.3) by scaling the genotype at each variant $i$ by its sample variance, i.e. $z_{ik} = \frac{x_{ik} - 2f_i}{\sqrt{\mathbb{V}(x_i)}}$ (Fig. 4c, Appendix). We can also recover term (1.4) by scaling the genotypes with the covariance between SNPs, i.e. the LD matrix, as previously proposed (Mathew *et al.* 2017; Ma and Dicker 2019) (Methods). In matrix form, the "LD-scaled" genotypes can be written as $\mathbf{Z} = (\mathbf{X} - 2\mathbf{P})\mathbf{U}^{-1}$ where $\mathbf{P}$ is an $n \times m$ matrix such that all elements of the $i$th column contain the frequency of the $i$th SNP and $\mathbf{U}$ is the (upper triangular) square root matrix of the LD matrix, i.e. $\mathbf{\Sigma} = \mathbf{U}'\mathbf{U}$ (Mathew *et al.* 2017). GREML recovers the LD contribution under this scaling, resulting in unbiased estimates of $V_g$ for both traits (Fig. 4d, Appendix).

In practice, however, the LD contribution may not be fully recoverable for two reasons. One, the LD-scaled GRM requires computing the inverse of $\mathbf{\Sigma}$ or $\mathbf{U}$ which may not exist, especially if the number of markers is greater than the sample size—the case for most human genetic studies. Second, it is common to include individual ancestry or principal components of the GRM as fixed effects in the model to account for inflation in heritability estimates due to shared environment. This also has the effect of removing the components of genetic variance along the ancestry axes, the residual variance being equal to $\mathbb{E}\{\mathbb{V}(g|\theta)\} = (1.1) + (1.2) - (1.3)1.11.21.3$ (Appendix). Indeed, this is what we observe in Fig. 4h.

## HE estimation

HE regression also assumes a random-effects model but uses a method-of-moments approach, as opposed to GREML, which maximizes the likelihood to estimate $V_g$. Previous work has shown that as long as all causal variants are included in the GRM calculation, the HE estimator will not be biased, even if they are in LD with each other (Min *et al.* 2022). We show that in the presence of positive and negative LD between causal loci, as exemplified by traits under divergent and stabilizing selection, respectively, the HE estimates of $V_g$ *are* biased upwards and downwards, respectively (Fig. 5a and b). To understand this discrepancy and the source of bias in our simulations, recall that HE estimates $V_g$ from the regression of the (pairwise) phenotypic covariance between individuals on their genotypic covariance (Haseman and Elston 1972). More specifically, if we denote $Y_{kl} = y_k y_l$ as the product of the (centered) phenotypes of $k$th and $l$th individuals, and $\psi_{kl}$ as the $k$th and $l$th entry of the GRM, then the HE estimator can be written as:

$$
\begin{aligned}
\hat{V}_g^{he} &= \frac{\mathrm{Cov}(Y_{kl}, \psi_{kl})}{\mathrm{Var}(\psi_{kl})} = \frac{\mathbb{E}(y_k y_l \sum_{w=1}^{M} z_{wk} z_{wl})}{\mathbb{E}(\sum_{i=1}^{M} z_{ik} z_{il} \sum_{w=1}^{M} z_{wk} z_{wl})} \\
&= \frac{\mathbb{E}\{(g_k + e_k)(g_l + e_l) \sum_{w=1}^{M} z_{wk} z_{wl}\}}{\mathbb{E}(\sum_{i=1}^{M} z_{ik} z_{il} \sum_{w=1}^{M} z_{wk} z_{wl})} = \frac{\mathbb{E}(g_k g_l \sum_{w=1}^{M} z_{wk} z_{wl})}{\mathbb{E}(\sum_{i=1}^{M} z_{ik} z_{il} \sum_{j=1}^{M} z_{wk} z_{wl})} \\
&= \frac{\mathbb{E}\left(\sum_{i=1}^{M} \sum_{j=1}^{M} u_i u_j z_{ik} z_{jl} \sum_{w=1}^{M} z_{wk} z_{wl}\right)}{\mathbb{E}(\sum_{i=1}^{M} z_{ik} z_{il} \sum_{w=1}^{M} z_{wk} z_{wl})} \\
&= \frac{\mathbb{E}\left(\sum_{i=1}^{M} \sum_{j=1}^{M} u_i u_j \sum_{w=1}^{M} z_{ik} z_{jl} z_{wk} z_{wl}\right)}{\mathbb{E}(\sum_{i=1}^{M} \sum_{w=1}^{M} z_{ik} z_{il} z_{wk} z_{wl})} \\
&= \underbrace{\frac{\mathbb{E}\left(\sum_{i=1}^{M} u_i^2 \sum_{w=1}^{M} z_{ik} z_{jl} z_{wk} z_{wl}\right)}{\mathbb{E}(\sum_{i=1}^{M} \sum_{w=1}^{M} z_{ik} z_{il} z_{wk} z_{wl})}}_{\text{genic component}} \\
&\quad + \underbrace{\frac{\mathbb{E}\left(\sum_{i=1}^{M} \sum_{j \neq i} u_i u_j \sum_{w=1}^{M} z_{ik} z_{jl} z_{wk} z_{wl}\right)}{\mathbb{E}\left(\sum_{i=1}^{M} \sum_{w=1}^{M} z_{ik} z_{il} z_{wk} z_{wl}\right)}}_{\text{directional LD}}
\end{aligned}
\tag{2}
$$

Where the first and second terms represent the genic and LD components, respectively, of the estimate. Population structure induces correlations between the alleles at a given locus as well as across loci (i.e. LD). But the LD may not be directional, i.e. trait-increasing alleles may be as likely to be co-inherited with each other as they are to be trait-decreasing alleles, and vice versa—implicit under the standard random-effects model. Thus, in the absence of directional LD, the second term is zero and the first term is unaffected as long as all causal variants are included in the GRM, because the increase in the numerator due to population structure is proportional to the denominator (Min *et al.* 2022). Directional LD does not affect the first term but exaggerates the contribution from the second term, i.e. the LD component (see Appendix section Directional LD). Consequently, HE regression over- and under-estimates $V_g$ for traits with positive and negative LD, respectively. Note that this bias is in the opposite direction of the bias observed with GREML, which fails to capture the LD contribution. Scaling the genotype at a locus by its LD with other loci, as discussed in the previous section, corrects for the bias in HE regression regardless of genetic architecture, yielding estimates consistent with GREML (Fig. 5c). Thus, GREML and HE regression are guaranteed to yield the same estimates only if the underlying model specifying the distribution of effects is consistent with the true architecture of the trait.

As with GREML, it is common to account for ancestry in HE regression to correct for any inflation in heritability estimates due to shared environment (Ge *et al.* 2017; Lin Z *et al.* 2022). We show that this removes the bias from exaggerated LD contributions but also removes any genetic variation along the ancestry axis (Fig. 5d–f, Methods). Thus, ancestry-corrected estimates of $V_g$ should be consistent between GREML and HE regression and should be interpreted as the sum of the contributions of individual loci, i.e. the genic variance.

## LD score regression

LD score regression (LDSC) is different from GREML and HE regression in that it uses GWAS summary statistics instead of individual-level data (Bulik-Sullivan *et al.* 2015). Intuitively, LDSC estimates $V_g$ (or $h_{snp}^2$ if the phenotype is scaled to unit variance) from the slope of a regression of marginal association statistics from GWAS ($\chi_k^2$) on "LD scores," defined as the sum of squared correlations between a given $k$th variant and all other variants, i.e. $l_k = \sum_{i=1}^{m} r_{ik}^2$. To understand the effect of directional LD on LDSC, note that the estimated marginal effect ($\hat{\tau}_k$) of the $k$th marker is a function of its true marginal effect ($\tau_k$) and some estimation error ($\varepsilon_k$): $\hat{\tau}_k = \tau_k + \varepsilon_k$ assuming for simplicity that there is no residual stratification in the upstream GWAS. Then, $\mathbb{E}(\hat{\tau}_k) = \tau_k$ and $\mathbb{E}(\hat{\tau}_k^2) = \tau_k^2 + \mathbb{E}(\varepsilon^2) = \tau_k^2 + SE^2(\hat{\tau}_k)$, where $SE(.)$ is the standard error. Furthermore, $\tau_k = \sum_{i=1}^{m} r_{ik} u_i$, where $r_{ik}$ is the genotypic correlation between the $k$th marker and $i$th causal variant (Bulik-Sullivan *et al.* 2015; Pirinen 2023).

$$
\mathbb{E}(\chi_k^2) = \frac{\mathbb{E}(\hat{\tau}_k^2)}{SE^2(\hat{\tau}_k)} = \frac{\tau_k^2 + SE^2(\hat{\tau}_k)}{SE^2(\hat{\tau}_k)}
$$

If we assume a highly polygenic architecture—as LDSC does—such that the effect of any individual causal variant is small, then $SE^2(\hat{\tau}_k) \approx \frac{1}{n\mathbb{V}(z_k)} = 1/n$ and $\mathbb{E}(\chi_k^2) \approx n\tau_k^2 + 1$. We can decompose the LDSC regression slope as follows:

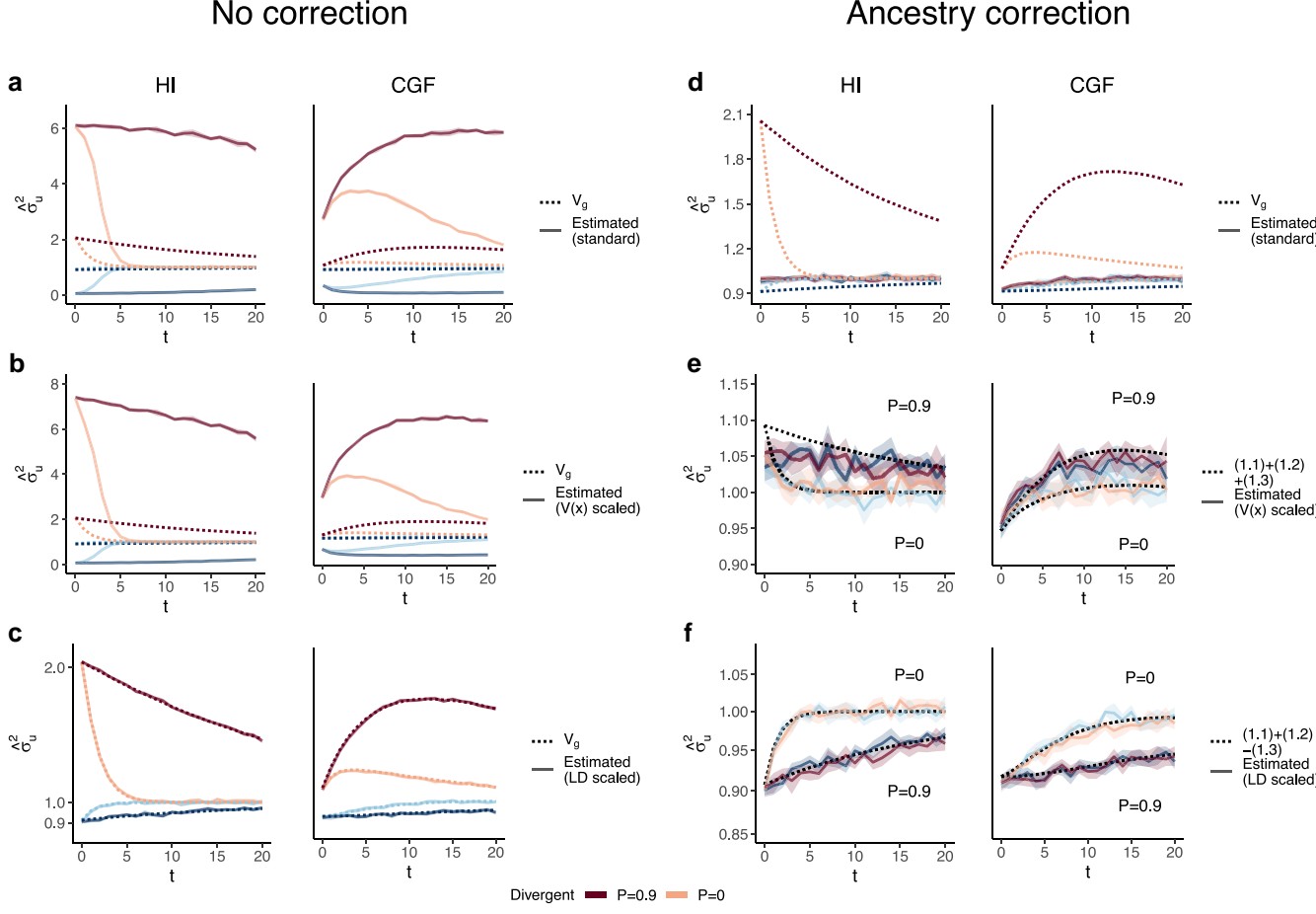

**Fig. 5.** Genetic variance ($\hat{V}_g$) estimated with HE regression in admixed populations under the HI (left column) and CGF (right column) models either without (a–c) or with (d–f) adjustment for individual ancestry. The solid lines represent estimates from simulated data averaged across 10 replicates. $P$ indicates the strength of assortative mating. (a and d) show the behavior of $\hat{V}_g$ for the default scaling, (b, e) shows $\hat{V}_g$ when the genotype at a locus is scaled by its sample variance ($\mathbb{V}(x)$ scaled), and (c, f) when it is scaled by the sample covariance (LD scaled). The dotted lines in (a–e) represent the expected $V_g$ in the population based on Equation 1 and in (f), represent the expected $V_g$ after removing any genetic variance along the ancestry axis. The shaded areas represent the 95% bootstrapped confidence bands of the estimate.

$$\mathbb{E}(\hat{\beta}_{\mathrm{ldsc}}) = \frac{\mathbb{C}(\chi_k^2, l_k)}{\mathbb{V}(l_k)} \approx \frac{\mathbb{C}(n\tau_k^2 + 1, l_k)}{\mathbb{V}(l_k)} = \frac{\mathbb{C}\{n(\sum_{i=1}^m r_{ik}u_i)^2, l_k\}}{\mathbb{V}(l_k)}$$

$$\approx \frac{n\mathbb{C}\left(\sum_{i=1}^m r_{ik}^2 u_i^2 + 2\sum_{i=1}\sum_{j<i}^m r_{ik}r_{jk}u_iu_j, l_k\right)}{\mathbb{V}(l_k)}$$

$$\approx \underbrace{\frac{n\mathbb{C}(\sum_{i=1}^m r_{ik}^2 u_i^2, l_k)}{\mathbb{V}(l_k)}}_{\text{genic component}} + \underbrace{\frac{n\mathbb{C}\left(2\sum_{i=1}^m \sum_{j<i} r_{ik}r_{jk}u_iu_j, l_k\right)}{\mathbb{V}(l_k)}}_{\text{directional LD}}$$

Thus, the LDSC slope can also be decomposed into contributions of individual loci (first term) and directional LD (second term). As with GREML and HE regression, LDSC assumes random independent effects where $u_i \sim \mathcal{N}(0, \frac{\sigma_u^2}{m})$ such that the second term is zero over variants $i$ and $j$, and the slope reduces to:

$$\mathbb{E}(\hat{\beta}_{\mathrm{ldsc}}) = \frac{n\mathbb{C}(\sum_{i=1}^m r_{ik}^2 u_i^2, l_k)}{\mathbb{V}(l_k)} = \frac{n\sigma_u^2}{m}\frac{\mathbb{C}(\sum_{i=1}^m r_{ik}^2, l_k)}{\mathbb{V}(l_k)} = \frac{n\sigma_u^2}{m}$$

from which $\hat{V}_g$ can be derived, i.e. $\hat{V}_g = \hat{\sigma}_u^2 = \frac{\hat{\beta}_{\mathrm{ldsc}}m}{n}$.

In practice, LD scores are computed in a user-defined window around each variant. If the causal variants are far enough apart, i.e. not in physical linkage—implicit in our generative model—such that there is effectively only a single variant per window, or if there is no directional LD within a window, the LDSC estimate of $V_g$ should reflect the sum of the contributions of individual loci, i.e. the genic variance, and therefore, be equivalent to (ancestry-adjusted) GREML and HE regression. As a result, LDSC estimates of $V_g$, and therefore, of $h_{\mathrm{snp}}^2$, will also be biased downwards and upwards in the presence of positive and negative LD, respectively, across causal variants.

To illustrate this, we simulated genotype data on chromosome 2 (M = 88,112 SNPs) of individuals of mixed African and European ancestry (N = 5,000) under the HI and CGF models using haplotypes from the 1000 Genomes YRI and CEU (Auton *et al.* 2015) (Methods). We selected $p \in \{0.01, 0.05, 0.1\}$ proportion of variants uniformly at random to be causal and assigned them effects $\beta_i = \frac{1}{\sqrt{pM2\bar{f}_i(1-\bar{f}_i)}}$, where $\bar{f}_i$ is the mean frequency between YRI and CEU. To simulate directional LD, we assigned "+" or "−" signs to causal effects uniformly at random 1,000 times and selected the

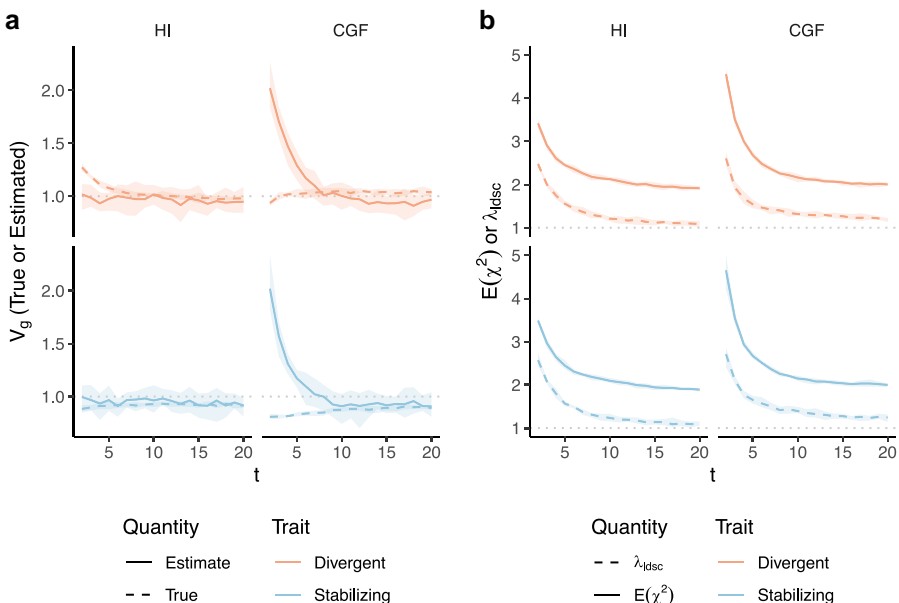

**Fig. 6.** Behavior of LDSC estimates of (a) $V_g$ and (b) $\mathbb{E}(\chi^2)$ under the HI and CGF admixture models for traits under divergent (top row) and stabilizing (bottom row) selection as a function of time since admixture (x-axis). In (a), the dashed and solid lines indicate the true (simulated) and estimated $V_g$, respectively, while the dotted horizontal line shows the expected genic variance. In (b), the solid line shows the mean $\chi^2$ of the GWAS summary statistics, while the dashed line shows the LDSC intercept ($\lambda_{\text{ldsc}}$). Shaded area indicates the 95% CI across 10 replicates. Results here shown only for $p = 0.01$. Results for $p \in \{0.05, 0.1\}$ are shown in Supplementary Fig. 3.

combination that gave the largest and smallest $V_{gb}$ between YRI and CEU to represent traits under divergent and stabilizing selection, respectively. We carried out GWAS for the unscaled genetic values in the admixed population at every generation under the HI and CGF models with 20 genetic principal components (PCs) as covariates. We also projected out 20 PCs from the genotypes in computing LD scores to account for admixture LD with cov-LDSC (Luo et al. 2021). Because we used the unscaled genetic values as our GWAS phenotype, the LDSC slope gives an estimate of $V_g$, the behavior of which is shown in Fig. 6 as a function of genetic architecture and admixture history.

Under the HI model, the LDSC $\hat{V}_g$ is also consistent with an estimate of the genic variance, in line with theoretical expectation. As a result, LDSC under- and over-estimates $V_g$, and therefore, $h^2_{\text{snp}}$ for traits under divergent and stabilizing selection, respectively (Fig. 6a). With time and random mating, $V_g$ will converge to the genic variance, and therefore, to $\hat{V}_g$. The pattern is more complex for the CGF model, which shows an inflation in $\hat{V}_g$ in the first few generations of admixture, regardless of genetic architecture (Fig. 6a). We believe this is due to inflation in the GWAS summary statistics that is not fully captured by the LDSC intercept ($\lambda_{\text{ldsc}}$). This can be seen in Fig. 6b, which shows that the mean $\chi^2$ is larger in the first few generations of the CGF model compared with that in the HI model even though the simulated effects are the same. We confirm this by observing over-dispersion in GWAS summary statistics especially in the first few generations of the CGF model (Supplementary Fig. 4). Importantly, the LDSC intercepts are similar under the HI and CGF models, suggesting that at least some of the inflation in summary statistics is not captured by $\lambda_{\text{ldsc}}$ under the CGF model. Because we performed GWAS on genetic values directly, the inflation in test statistics is not because of environmental stratification. Instead, we believe it arises from subtle patterns of admixture LD in the first few generations of the CGF model when the minor (CEU) ancestry is rare in the sample. As such, rare variants tagging CEU ancestry segments would be

susceptible to subtle biases that are difficult to correct by including PCs as linear covariates in the GWAS (Supplementary Fig. 4e). The residual inflation, because it might be correlated with admixture LD, is absorbed by the LDSC slope, leading to an overestimate in $V_g$, regardless of genetic architecture. Consistent with this, the bias appears to decrease with time under the CGF model (Fig. 6) as the proportion of CEU ancestry increases in the population. As the bias decreases, the LDSC estimate converges to the genic variance. Importantly, this estimate is also consistent with (ancestry-adjusted) estimates from GREML and HE regression. We confirm this by applying GREML and HE regression to data from the same set of simulations used for LDSC (Supplementary Fig. 5).

In summary, LDSC estimates of $V_g$ and $h^2_{\text{snp}}$ are susceptible to confounding due to population structure in admixed populations if the minor ancestry is represented by a small proportion of individuals in the GWAS sample. But as long as this is not the case, LDSC estimates of $V_g$ and $h^2_{\text{snp}}$ should also be interpreted as the sum of the contributions of individual loci, and therefore should be consistent with GREML and HE regression. This estimate may or may not be equal to $h^2$, depending on the genetic architecture.

## Local ancestry heritability

A related quantity of interest in admixed populations is local ancestry heritability ($h^2_\gamma$), which is defined as the proportion of phenotypic variance that can be explained by local ancestry. Zaitlen et al. (2014) showed that this quantity is related to, and can be used to estimate, $h^2$ in admixed populations. The advantage of this approach is that local ancestry segments shared between individuals are identical by descent and are therefore more likely to tag causal variants compared with array markers, allowing one to potentially capture the contributions of rare variants (Zaitlen et al. 2014). Here, we show that in the presence of population structure, (1) the relationship between $h^2_\gamma$ and $h^2$ is

not straightforward and (3) $\hat{h}_\gamma^2$ may be a biased estimate of local ancestry heritability under the random effects model for the same reasons that $\hat{h}_{snp}^2$ is biased.

We define local ancestry $\gamma_i \in \{0, 1, 2\}$ as the number of alleles at locus i that trace their ancestry to population A. Thus, ancestry at the ith locus in individual $k$ is a binomial random variable with $\mathbb{E}(\gamma_{ik}) = 2\theta_k$, $\theta_k$ being the ancestry of the kth individual. Similar to genetic value, the "ancestry value" of an individual can be defined as $\sum_{i=1}^{m} \phi_i \gamma_i$, where $\phi_i = \beta_i(f_i^A - f_i^B)$ is the effect size of local ancestry (Appendix). Then, the genetic variance due to local ancestry can be expressed as (Appendix):

$$
\begin{aligned}
V_\gamma &= \mathbb{V}\left(\sum_{i=1}^{m} \phi_i \gamma_i\right) = \sum_{i=1}^{m} \phi_i^2 \mathbb{V}(\gamma_i) + \sum_{i=1}^{m}\sum_{j \neq i}^{m} \phi_i \phi_j \mathbb{C}(\gamma_i, \gamma_j) \\
&= 2\mathbb{E}(\theta)\{1 - \mathbb{E}(\theta)\}\sum_{i=1}^{m} \phi_i^2 + 2\mathbb{V}(\theta)\sum_{i=1}^{m} \phi_i^2 + 4\mathbb{V}(\theta)\sum_{i=1}^{m}\sum_{j \neq i}^{m} \phi_i \phi_j \\
&= 2\mathbb{E}(\theta)\{1 - \mathbb{E}(\theta)\}\sum_{i=1}^{m} \beta_i^2(f_i^A - f_i^B)^2 \\
&\quad + 2\mathbb{V}(\theta)\sum_{i=1}^{m} \beta_i^2(f_i^A - f_i^B)^2 \\
&\quad + 4\mathbb{V}(\theta)\sum_{i=1}^{m}\sum_{j \neq i}^{m} \beta_i \beta_j (f_i^A - f_i^B)(f_j^A - f_j^B)
\end{aligned}
$$

and heritability explained by local ancestry is simply the ratio of $V_\gamma$ and the phenotypic variance. Note that $V_\gamma = (1.2) + (1.3) + (1.4)$1.21.31.4 and therefore its behavior is similar to $V_g$ in that the terms (1.3) and (1.4) decay towards zero as $\mathbb{V}(\theta) \to 0$, and $V_\gamma$ converges to (1.2) (Supplementary Fig. 6). Additionally, the dependence of $V_\gamma$ on both $\mathbb{E}(\theta)$ and $\mathbb{V}(\theta)$ precludes a straightforward derivation between local ancestry heritability and $h^2$.

GREML estimation of $\hat{h}_\gamma^2$ is similar to that of $\hat{h}_{snp}^2$, the key difference being that the former involves constructing the GRM using local ancestry instead of genotypes (Zaitlen *et al.* 2014). The following model is assumed: $\mathbf{y} = \mathbf{W}\boldsymbol{\upsilon} + \boldsymbol{\xi}$, where $\mathbf{W}$ is an $n \times m$ standardized local ancestry matrix, $\boldsymbol{\upsilon} \sim \mathcal{N}(0, \mathbf{I}\frac{\sigma_\upsilon^2}{m})$ are local ancestry effects, and $\boldsymbol{\xi} \sim \mathcal{N}(0, \mathbf{I}\sigma_\xi^2)$. Note that $\sigma_\xi^2$ captures both environmental noise as well as any genetic variance independent of local ancestry. The phenotypic variance is decomposed as $\mathbb{V}(\mathbf{y}) = \mathbb{V}(\mathbf{W}\boldsymbol{\upsilon}) + \mathbb{V}(\boldsymbol{\xi}) = \frac{\mathbf{W}\mathbf{W}'}{m}\sigma_\upsilon^2 + \sigma_\xi^2$ where $\frac{\mathbf{W}\mathbf{W}'}{m}$ is the local ancestry GRM and $\sigma_\upsilon^2$ is the parameter of interest, which is believed to be equal to $V_\gamma$—the genetic variance due to local ancestry.

We show that, in the presence of population structure, i.e. when $\mathbb{V}(\theta) > 0$, GREML $\hat{\sigma}_\upsilon^2$ is biased downwards relative to $V_\gamma$ for traits under divergent selection and upwards for traits under stabilizing selection because it does not capture the contribution of LD (Fig. 7a). But there is another source of bias in $\hat{\sigma}_\upsilon^2$, which tends to be inflated in the presence of population structure if individual ancestry is not included as a covariate, even with respect to the expectation of $V_\gamma$ under equilibrium (seen more clearly in Fig. 7b and c). We suspect this inflation is because of strong correlations between local ancestry—local ancestry disequilibrium—across loci that inflates $\hat{\sigma}_\upsilon^2$ in a way that is not adequately corrected even when all causal variants are included in the model (Yang, Manolio, *et al.* 2011; Yang *et al.* 2016). Scaling local ancestry by its covariance removes this bias and recovers the contribution of LD (Fig. 7d) presumably because this accounts for the correlation in genotypes across loci. Including individual ancestry as a fixed effect also corrects for the inflation in $\hat{\sigma}_\upsilon^2$ (Fig. 7e–h). But as with $\hat{\sigma}_u^2$, this practice will underestimate the genetic variance due to

local ancestry in the presence of population structure because it removes the variance along the ancestry axis (Fig. 7e–h).

Based on the above, GREML $\hat{h}_\gamma^2$ and corresponding estimates of $h^2$ are more accurately interpreted as the heritability due to local ancestry and heritability, respectively, expected in the absence of population structure. We believe $\hat{h}_\gamma^2$ is still useful in that, because it should capture the effects of rare variants, it can be used to estimate the upper bound of $\hat{h}_{snp}^2$.

In a previous paper, we suggested that local ancestry heritability could potentially be used to estimate the genetic variance between populations (Zaidi *et al.* 2017). Our results suggest this is not possible for two reasons. First, the GREML estimator of local ancestry heritability, as we show in this section is biased and does not capture the LD contribution. But even if we were able to recover the LD component, our decomposition shows that local ancestry is equal to the genetic variance between populations ($V_{gb}$) only when $\mathbb{E}(\theta) = 0.5$ and $\mathbb{V}(\theta) = \mathbb{E}(\theta)\{1 - \mathbb{E}(\theta)\} = 0.25$, which is only possible at $t = 0$ in the HI model. After admixture, $\mathbb{V}(\theta)$ decays and the equivalence between $V_\gamma$ and $V_{gb}$ is lost, making it impossible to estimate the latter from admixed populations, especially for traits under divergent or stabilizing selection, even if the environment is randomly distributed with respect to ancestry. We note that this conclusion was recently reached independently by Schraiber and Edge (2024).

## How much does LD contribute to $V_g$ in practice?

In the previous sections, we showed theoretically that $\hat{h}_{snp}^2$ may be biased in admixed populations even in the absence of confounding by shared environment and even if the causal variants are directly genotyped. All three estimators fail to capture the LD contribution. The extent to which $\hat{h}_{snp}^2$ is biased because of this reason in practice is ultimately an empirical question, which is difficult to answer because the true genetic architecture—the LD contribution in particular—is unknown. In this section, we develop some intuition for this contribution among individuals with mixed African and European ancestry using a combination of simulations and empirical data analysis.

First, we simulated a neutral trait using genotype data from the African Americans (ASW) from the 1000 Genomes Project (Auton *et al.* 2015). To do this, we sampled $m \in \{10, 100, 1,000\}$ causal loci from a set of common (MAF >0.01), LD pruned variants and assigned them effects such that $\beta_i \sim \mathcal{N}(0, \frac{1}{\sqrt{m\mathbb{V}(x_i)}})$, i.e. the expected *genic* variance is $\mathbb{E}\{\sum_{i=1}^{m} \beta_i^2 \text{Var}(x_i)\} = 1$ (Methods). We computed the genic and LD contributions and repeated this process 1,000 times where each replicate can be thought of as an independent realization of the genetic architecture of a neutrally evolving trait. We show that the LD contribution may be zero in expectation but can be substantial for a given trait (up to 50% of the genic variance, Supplementary Fig. 7), even in the absence of selection.

Second, we estimated the LD contribution of genome-wide significant SNPs for 26 quantitative traits among admixed Americans with primarily African and European ancestry. To do this, we decomposed the variance explained by GWAS SNPs into the four components in Equation 1 using previously published effect sizes, allele frequencies ($f^A$ and $f^B$) from the 1000 Genomes YRI and CEU, and the mean ($\mathbb{E}(\theta) \approx 0.77$) and variance ($\mathbb{V}(\theta) \approx 0.02$) of individual ancestry from ASW (Methods). We show that for skin pigmentation—a trait under strong divergent selection—the LD contribution, i.e. term (1.4), is positive and accounts for $\approx 40\%$ of the total genetic variance. This is because of large allele frequency differences between Africans and Europeans that are correlated

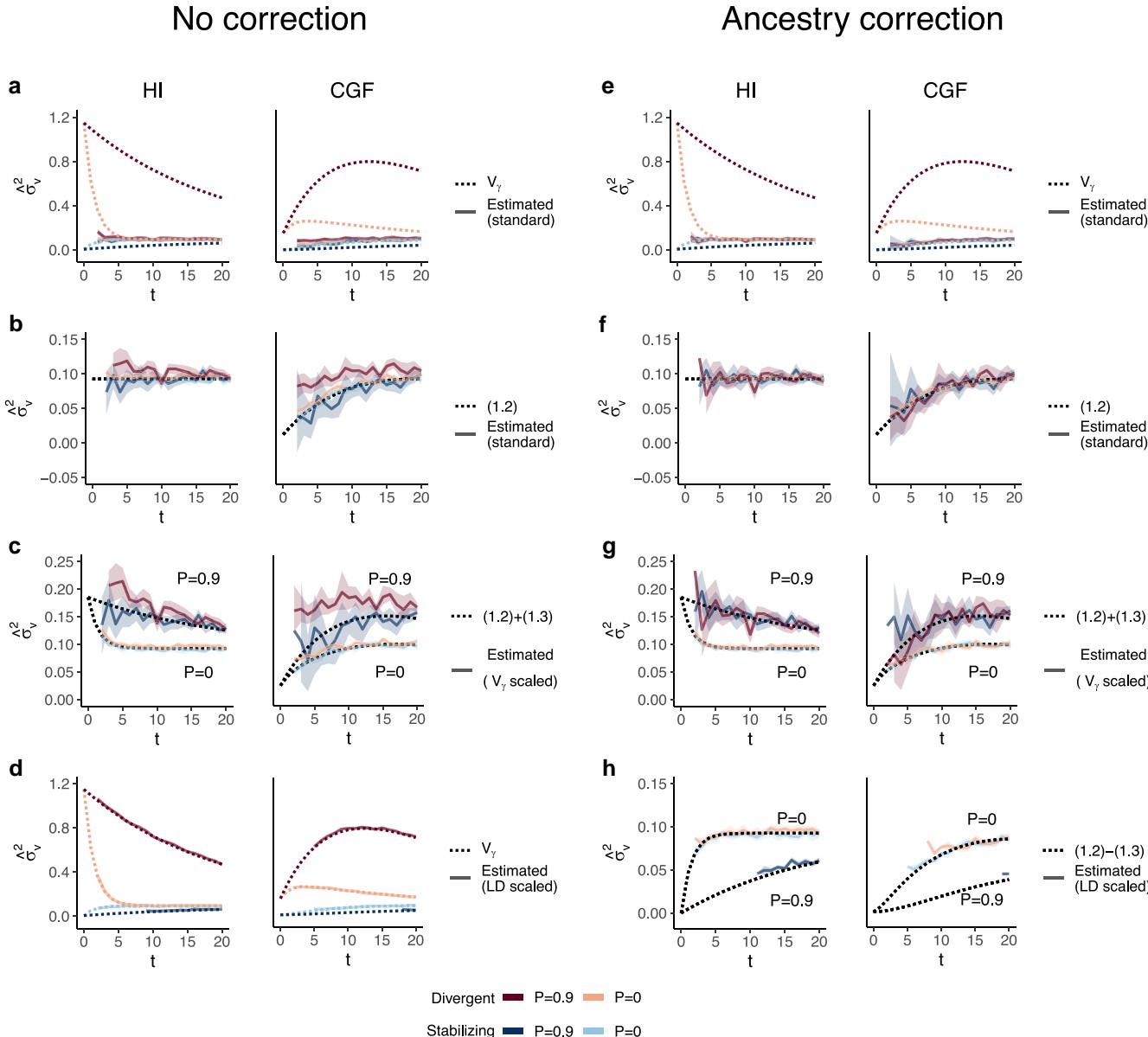

**Fig. 7.** The behavior of GREML estimates of the variance due to local ancestry ($\hat{\sigma}_v^2$) in admixed populations under the HI (left column) and CGF (right column) models either without (a–d) or with (e–h) individual ancestry included as a fixed effect. The solid lines represent estimates from simulated data averaged across 10 replicates. P indicates the strength of assortative mating. The dotted lines either represent the expected variance in the population (a and b) or the expected estimate for three different ways of scaling local ancestry (b–d and f–h). (a–b and e–f) show the behavior of $\hat{\sigma}_v^2$ for the default scaling, (c, g) shows $\hat{\sigma}_v^2$ when local ancestry is scaled by the sample variance, and (d, h) when it is scaled by the sample covariance. Shaded regions represent the 95% confidence bands. Some runs in (d and h) failed to converge as seen by the missing segments of the solid lines because the expected variance in such cases was too small.

across skin pigmentation loci, consistent with strong polygenic selection favoring alleles for darker pigmentation in regions with high UV exposure and vice versa (Jablonski 2004; Lamason *et al.* 2005; Beleza *et al.* 2013; Zaidi *et al.* 2017; Ju and Mathieson 2020). But for most other traits, LD contributes relatively little, explaining a modest, but nonnegligible proportion of the genetic variance. For example, GWAS SNPs for height, LDL and HDL cholesterol, and mean corpuscular hemoglobin (MCH) exhibit positive LD, whereas SNPs for blood count—particularly neutrophil (NEU) and white blood cell count (WBC)—loci exhibit negative LD (Fig. 8). Because we selected independent associations for this analysis (Methods), the LD contribution is driven largely due to population structure in ASW. The contribution of population

structure to the genic variance, i.e. term (1.3) is small even for traits like skin pigmentation and neutrophil count with large effect alleles that are highly diverged in frequency between Africans and Europeans (Lamason *et al.* 2005; Nalls *et al.* 2008; Reich *et al.* 2009; Beleza *et al.* 2013; McManus *et al.* 2017). Overall, this suggests a modest contribution of population structure to the genetic variance explained by GWAS SNPs for most traits. Nevertheless, this contribution is nonnegligible and may be higher for other traits or for variants that do not reach genome-wide significance. We recommend that, in addition to considering the contribution of population structure in overall heritability estimation, researchers also account for it when reporting the total variance explained by GWAS SNPs in admixed and other structured

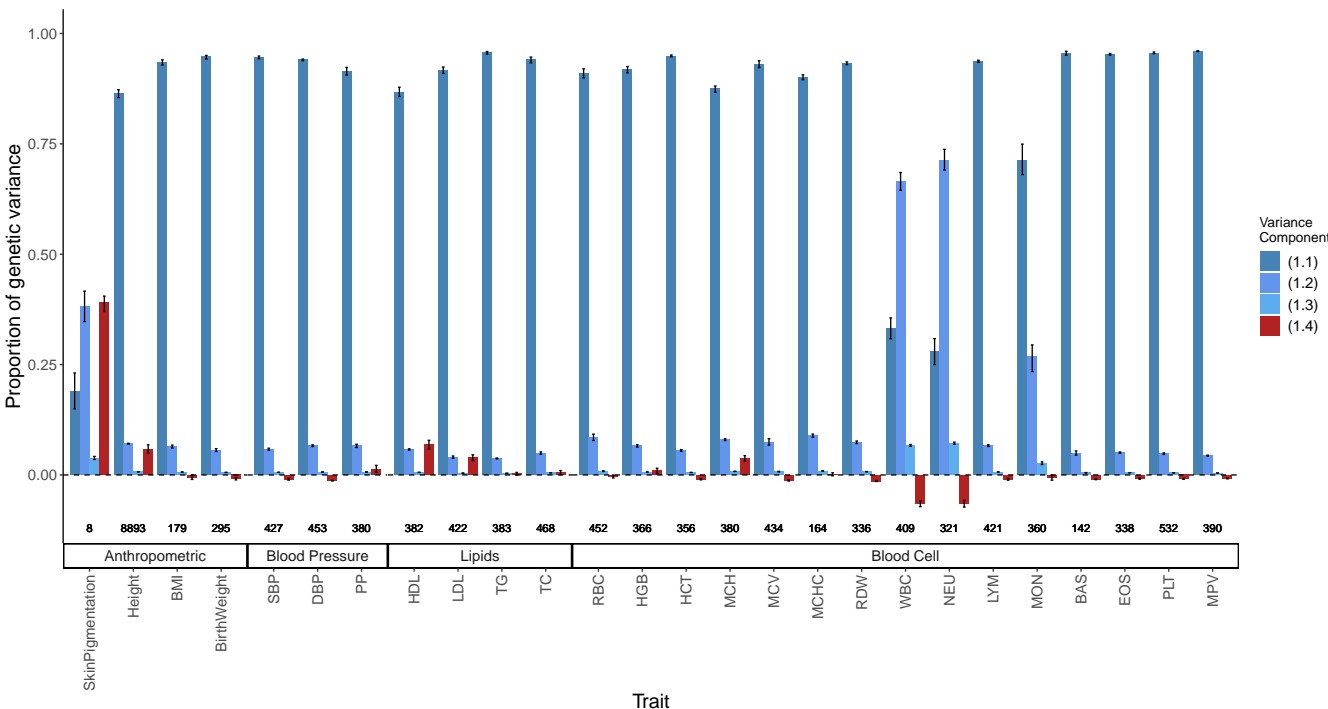

**Fig. 8.** Decomposing the genetic variance explained by GWAS SNPs in the 1000 Genomes ASW (African Americans from Southwest). We calculated the four variance components listed in Equation 1, their values shown on the y-axis as a proportion of total genetic variance and confidence intervals from a parametric bootstrap (Methods). The number of variants used to calculate variance components for each trait is shown at the bottom.

populations. This can be done with Equation 1 instead of the standard $2\sum_{i=1}^{m} \hat{\beta}_i^2 f_i (1 - f_i)$ formula used in most studies.

## Discussion

Despite the growing size of GWAS and discovery of thousands of variants for hundreds of traits (Sollis *et al.* 2023), the heritability explained by GWAS SNPs remains a fraction of twin-based heritability estimates. Yang *et al.* (2010) introduced the concept of SNP heritability ($h_{snp}^2$) that does not depend on the discovery of causal variants but assumes that they are numerous and are more or less uniformly distributed across the genome (the infinitesimal model), their contributions to the genetic variance "tagged" by genotyped SNPs (Yang *et al.* 2010). $h_{snp}^2$ is now routinely estimated in most genomic studies and at least for some traits (e.g. height and BMI), these estimates now approach twin-based heritability (Wainschtein *et al.* 2022). But despite the widespread use of $\hat{h}_{snp}^2$, its interpretation remains unclear, particularly in the presence of admixture and population structure. It is generally accepted that $\hat{h}_{snp}^2$ can be biased in structured populations because of confounding effects of unobserved environmental factors and LD between causal variants (Browning SR and Browning 2011; Yang, Manolio, *et al.* 2011; Kumar *et al.* 2016a, 2016b; Yang *et al.* 2016; Lin Z *et al.* 2022). But $\hat{h}_{snp}^2$ may be biased even in the absence of confounding because of misspecification of the underlying random-effects model, i.e. if the model does not represent the genetic architecture from which the trait is sampled (Speed *et al.* 2012, 2017; de los Campos *et al.* 2015; Rawlik *et al.* 2020; Lara *et al.* 2021).

Under the standard random-effects model, SNP effects are assumed to be uncorrelated and the total genetic variance can be represented as the sum of the variance explained by individual loci, i.e. the genic variance (de los Campos *et al.* 2015; Rawlik *et al.* 2020; Lara *et al.* 2021). In admixed populations, there is substantial LD, which can contribute to the genetic variance, and can persist for a number of generations, despite recombination, due to continued gene flow and/or ancestry-based assortative mating. GREML and HE regression do not capture this LD contribution, and therefore, may lead to biased estimates of $h_{snp}^2$. The LD contribution can be negative for traits under stabilizing selection, and positive for traits under divergent selection between the source populations, leading to over- or under-estimates, respectively. We show that in admixed and other structured populations, GREML and HE regression estimates can be biased even when the LD contribution is zero if the genotypes are scaled by $\sqrt{2f(1-f)}$—the standard approach, which implicitly assumes a randomly mating population. In the presence of population structure, the variance in genotypes can be higher and $\hat{h}_{snp}^2$ does not capture this additional variance, which we show can be recovered by scaling genotypes by the SNP variance ($\sqrt{\text{Var}(x)}$). Thus, as long as the genotypes are scaled properly GREML and HE estimates of $h_{snp}^2$ should be interpreted as the proportion of phenotypic variance explained by the *genic* variance. Estimates of local ancestry heritability ($\hat{h}_\gamma^2$) (Zaitlen *et al.* 2014; Chan *et al.* 2023) should be interpreted similarly.

The behavior of LD Score regression, which is different from GREML and HE in that it uses variant summary statistics as input as opposed to individual genotypes, depends on the magnitude and pattern of inflation in the GWAS and the extent to which it is captured by the LDSC intercept. We show that under the HI model, where all the admixture occurs at once, the LDSC estimate of $h_{snp}^2$ is equivalent to GREML and HE regression in that it

measures the sum of the individual contribution of each locus, i.e. the genic variance. However, we note that under some population structures, e.g. under the CGF model where the amount of gene flow in the first few generations is small, inflation in summary statistics due to population structure may not uniformly affect all variants and can be correlated with local levels of LD. This can lead to inflated estimates of $h_{\text{snp}}^2$ that are not interpretable. As such, we recommend caution when using LDSC in estimating heritability from GWAS in admixed and other structured cohorts, especially when the ancestry composition of the cohort is highly unbalanced, i.e. when certain ancestries are represented by a small number of individuals. We suggest that researchers remove any individuals who may represent a very small fraction of the GWAS sample (in terms of ancestry) as well as report SNP heritability estimates from all three approaches so that any inconsistencies can be evaluated for potential sources of bias.

Does the bias in SNP heritability estimates due to the missing LD contribution have practical implications? The answer to this depends on the context in which SNP heritability is used. In the absence of residual bias due to stratification (see previous paragraph), $\hat{h}_{\text{snp}}^2$ is an interpretable quantity: the sum of the proportion of phenotypic variance explained by individual loci. As such, $\hat{h}_{\text{snp}}^2$ can be useful in quantifying the power to detect variants in GWAS where the quantity of interest is the genic variance. But because it does not capture the LD contribution to $h^2$, $\hat{h}_{\text{snp}}^2$ may not be appropriate in measuring the extent to which genetic variation contributes to phenotypic variation, in predicting the response to selection, or in defining the upper limit of polygenic prediction accuracy (Visscher et al. 2008), especially for traits where the genetic risk varies as a function of ancestry (e.g. health disparities).

One limitation of this paper is that we have focused on random-effects estimators of $h_{\text{snp}}^2$ because of their widespread use. Estimators of $h_{\text{snp}}^2$ can be broadly grouped into random- and fixed effect estimators based on how they treat SNP effects (Min et al. 2022). Fixed effect estimators make fewer distributional assumptions but they are not as widely used because they require conditional estimates of all variants—a high-dimensional problem where the number of markers is often far larger than the sample size (Schwartzman et al. 2019). This is one reason why random effect estimators, such as GREML, are popular—because they reduce the number of parameters that need to be estimated by assuming that the effects are drawn from some distribution where the variance is the only parameter of interest. Fixed effects estimators, in principle, should be able to capture the LD contribution but this is not obvious in practice since the simulations used to evaluate the accuracy of such estimators still assume uncorrelated effects (Schwartzman et al. 2019; Min et al. 2022; Hou et al. 2023). Further research is needed to clarify the interpretation of the different estimators of $h_{\text{snp}}^2$ in structured populations under a range of genetic architectures.

Ultimately, the discrepancy between $\hat{h}_{\text{snp}}^2$ and $h^2$ in practice is an empirical question, the answer to which depends on the degree of population structure (which we can measure) and the genetic architecture of the trait (which we do not know a priori). We show that for most traits, the contribution of population structure to the variance explained by GWAS SNPs is modest among African Americans. Thus, if we assume that the genetic architecture of GWAS SNPs represents that of all causal variants, then despite incorrect assumptions, the discrepancy between $\hat{h}_{\text{snp}}^2$ and $h^2$ should be fairly small. But this assumption is unrealistic given that GWAS SNPs are common variants that in most cases cumulatively explain a fraction of trait heritability. What is the LD contribution of the rest of the genome, particularly rare variants? This is not obvious and will become clearer in the near future through large sequence-based studies (Backman et al. 2021). While these are underway, theoretical studies are needed to understand how different selection regimes influence the directional LD between causal variants—clearly an important aspect of the genetic architecture of complex traits.

## Materials and Methods
### Simulating genetic architecture

We first drew the allele frequency ($f^0$) of 1,000 biallelic causal loci in the ancestor of populations A and B from a uniform distribution, $U(0.001, 0.999)$. Then, we simulated their frequency in populations A and B ($f^A$ and $f^B$) under the Balding–Nichols model (Balding and Nichols 1995), such that $f^A$, $f^B \sim \text{Beta}(\frac{f^0(1-F)}{F}, \frac{(1-f^0)(1-F)}{F})$, where $F = 0.2$ is the inbreeding coefficient. We implemented this using code adapted from Lin M et al. (2021). To avoid drawing extremely rare alleles, we continued to draw $f^A$ and $f^B$ until we had 1,000 loci with $f^A$, $f^B \in (0.01, 0.99)$.

We generated the effect size of the $i$th locus by setting $\beta_i = \frac{1}{\sqrt{2m\bar{f}_i(1-\bar{f}_i)}}$, where $m$ is the number of loci and $\bar{f}$ is the mean allele frequency across populations A and B. Thus, rare variants have larger effects than common variants and the total genetic variance sums to 1. Given these effects, we simulated two different traits, one with a large difference in means between populations A and B (trait 1) and the other with roughly no difference (trait 2). This was achieved by permuting the signs of the effects 100 times to get a distribution of $V_{gb}$—the genetic variance between populations. This has the effect of varying the LD contribution without changing the $F_{ST}$ at causal loci. We selected the maximum and minimum of $V_{gb}$ to represent traits 1 and 2.

### Simulating admixture

We simulated the genotypes, local ancestry, and phenotype for 10,000 admixed individuals per generation under the hybrid isolation (HI) and continuous gene flow (CGF) models by adapting the code from Zaitlen et al. (2017). We denote the ancestry of a randomly selected individual $k$ with $\theta$, the fraction of their genome from population A. At $t = 0$ under the HI model, we set $\theta$ to 1 for individuals from population A and 0 if they were from population B such that $\mathbb{E}(\theta) = p \in \{0.1, 0.2, 0.5\}$ with no further gene flow from either source population. In the CGF model, population B receives a constant amount $q$ from population A in every generation starting at $t = 0$. The mean overall proportion of ancestry in the population is kept the same as the HI model by setting $q = 1 - (1 - p)^{\frac{1}{t}}$ where $t$ is the number of generations of gene flow from A. In every generation, we simulated ancestry-based assortative mating by selecting mates such that the correlation between their ancestries is $P \in \{0, 0.3, 0.6, 0.9\}$ in every generation. We do this by repeatedly permuting individuals with respect to each other until $P$ falls within $\pm 0.01$ of the desired value. It becomes difficult to meet this criterion when $\mathbb{V}(\theta)$ is small (Fig. 1c). To overcome this, we relaxed the threshold up to 0.04 for some conditions, i.e. when $\theta \in \{0.1, 0.2\}$ and $t \geq 50$. We generated expected variance in individual ancestry using the expression in Zaitlen et al. (2017). At time $t$ since admixture, $\mathbb{V}(\theta_t) = \mathbb{V}(\theta_{t-1})\frac{(1+P)}{2}$ under the HI model where $P$ measures the strength of assortative mating, i.e. the correlation between the ancestry between individuals in a mating pair.

Under the CGF model, $\mathbb{V}(\theta_t) = q(1-q)\mathbb{E}(\theta_{t-1})^2 + q(1-q)\{1 - 2\mathbb{E}(\theta_{t-1})\} + (1-q)\mathbb{V}(\theta_{t-1})\frac{(1+P)}{2}$ (Appendix).

We sampled the local ancestry at each ith locus as $\gamma_i = \gamma_{if} + \gamma_{im}$ where $\gamma_{im} \sim \text{Bin}(1, \theta_m)$, $\gamma_{if} \sim \text{Bin}(1, \theta_f)$ and $\theta_m$ and $\theta_f$ represent the ancestry of the maternal and paternal chromosome, respectively. The global ancestry of the individual is then calculated as $\theta_k = \sum_{i=1}^m \frac{\gamma_{im}+\gamma_{if}}{2m}$, where $m$ is the number of loci. We sample the genotypes $x_{im}$ and $x_{if}$ from a binomial distribution conditioning on local ancestry. For example, the genotype on the maternal chromosome is $x_{im} \sim \text{Bin}(1, f_i^A)$ if $\gamma_{im} = 1$ and $x_{im} \sim \text{Bin}(1, f_i^B)$ if $\gamma_{im} = 0$ where $f_i^A$ and $f_i^B$ represent the allele frequency in populations A and B, respectively. Then, the genotype can be obtained as the sum of the maternal and paternal genotypes: $x_i = x_{im} + x_{ip}$. We calculate the genetic value of each individual as $g = \sum_{i=1}^m \beta_i x_i$ and the genetic variance as $\mathbb{V}(g)$.

For LDSC estimation, we simulated phased genotypes on chromosome 2 ($M = 88,112$ HapMap 3 SNPs) for $N = 5,000$ individuals for 20 generations of admixture under the HI and CGF models from CEU and YRI haplotypes from the 1000 Genomes Project (Auton *et al.* 2015) using Haptools (Massarat *et al.* 2023) implemented in Admix-kit (Hou *et al.* 2024). We used recombination maps for the hg38 build of the Human genome downloadable from the Beagle (v5.5) website (Browning BL *et al.* 2018). We sampled $p \in \{0.01, 0.05, 0.1\}$ of the variants to be causal and assigned them effects such that $\beta_i = \frac{1}{2pM\bar{f}(1-\bar{f})}$ where $\bar{f} = \frac{f_{CEU}+f_{YRI}}{2}$ with the condition that $0.01 < \bar{f} < 0.99$. We permuted the signs of the effects 1,000 times and selected the combination that gave the smallest and largest value of genetic variance between CEU and YRI ($V_{gb}$) to represent traits under stabilizing and divergent selection, respectively.

## Heritability estimation with GREML

We used the --*reml* and --*reml-no-constrain* flags in GCTA (Yang, Lee, *et al.* 2011) to estimate $\sigma_u^2$ and $\sigma_v^2$, the genetic variance due to genotypes and local ancestry, respectively. We could not run GCTA without noise in the genetic values so we simulated individual phenotypes with a heritability of $h^2 = 0.8$ by adding random noise $e \sim \mathcal{N}(0, V_g \frac{1-h^2}{h^2})$. We computed three different GRMs, which correspond to different transformations of the genotypes: (1) standard, (2) variance or $V(x)$ scaled, and (3) LD-scaled.

For the standard GRM, the genotypes at the ith SNP are standardized such that $z_i = \frac{x_i - 2f_i}{\sqrt{2f_i(1-f_i)}}$. For the variance scaled GRM, we computed $z_i = \frac{x_i - 2f_i}{\sqrt{\mathbb{V}(x_i)}}$ where $\mathbb{V}(x_i)$ is the sample variance of the genotypes at the ith SNP. The LD-scaled GRM conceptually corresponds to standardizing the genotypes by the SNP covariance, rather than its variance. Let $\boldsymbol{X}$ represent the $n \times m$ *unstandardized* matrix of genotypes and $\boldsymbol{P}$ represent an $n \times m$ matrix where the ith column contains the allele frequency of that SNP. Let $\boldsymbol{U}$ be the upper triangular "square root" matrix of the $m \times m$ SNP covariance matrix $\boldsymbol{\Sigma}$ such that $\boldsymbol{\Sigma} = \boldsymbol{U'U}$. Then, the standardized genotypes are computed as $\boldsymbol{Z} = (\boldsymbol{X} - 2\boldsymbol{P})\boldsymbol{U}^{-1}$ and the GRM becomes $(\boldsymbol{X} - 2\boldsymbol{P})\boldsymbol{\Sigma}^{-1}(\boldsymbol{X} - 2\boldsymbol{P})'$ (Mathew *et al.* 2017). Similarly, the three GRMs for local ancestry were computed by scaling local ancestry with (1) $\sqrt{2\bar{\gamma}_i(1-\bar{\gamma}_i)}$ where we denote $\bar{\gamma}_i$ as the mean local ancestry at the ith SNP, or with (2) variance, or (3) covariance of local ancestry, respectively. We estimated $\sigma_u^2$ and $\sigma_v^2$ with and without individual ancestry as a fixed effect to correct for any confounding due to genetic stratification. This was done by using the --*qcovar* flag.

## Heritability estimation with HE regression

Haseman–Elston regression without ancestry correction was implemented using GCTA (Yang, Manolio, *et al.* 2011). To demonstrate that the bias in unadjusted HE estimates arises because of a bias in the estimate of LD contribution, not the genic variance, we carried out a simple simulation where half of the individuals in the population derive their ancestry from population A and the rest from population B. This is equivalent to the meta-population under the HI model at $t = 0$ where $\mathbb{E}(\theta) = 0.5$. We simulated genotypes for 1,000 individuals for $m = 100$ loci where the allele frequencies in populations A and B were set to $f_A = 0.1$ and $f_B = 0.8$, respectively. We standardized the genotypes at each locus $i$ using the square-root of the sample variance and assigned effect sizes such that the total genetic variance explained by all loci is equal to 1, i.e. the effect of the scaled genotype at the ith locus is $u_i = \frac{1}{\sqrt{m}}$. This is equivalent to the effect size of the unscaled genotypes being $\beta_i = \frac{1}{\sqrt{m\mathbb{V}(x_i)}}$ where $\mathbb{V}(x_i)$ is the sample variance at the ith locus. We introduced randomness in the direction of the effect by assigning a negative or positive sign to each locus uniformly at random 100 times to generate 100 traits with the same genic variance but varying LD contributions. Then, for each trait we computed the two terms in Equation 2, which should converge to the genic variance and LD contributions, which represent the genic and LD components to the HE regression estimate. Supplementary Fig. 8 shows that in the presence of directional LD, the overall bias is in the HE regression estimate is due to an exaggerated estimate of the LD contribution.

To correct for individual ancestry, we followed the moment-matching approach of Ge *et al.* (2017) implemented in MMHE. In this approach, conceptually, individual ancestry is first projected out from both the phenotype and genotypes. Then, the cross-product of the residual phenotypes is regressed on the corresponding entries of the GRM (computed from the residual genotypes) to estimate $V_g$. In practice, a more efficient algorithm is used.

## Heritability estimation with LDSC

We used estimate LD score regression heritability with cov-LDSC (Luo *et al.* 2021). We used summary statistics from a GWAS of the genetic values with 20 PCs as covariates. We used cov-LDSC to compute ancestry-adjusted LD scores (for 20 PCs) and to estimate the LDSC slope and intercept ($\lambda_{ldsc}$) for each combination of the parameter space, i.e. (1) demographic model (HI or CGF), (2) generations since admixture (2–20), (3) genetic architecture (divergent or stabilizing), (4) proportion of variants that are causal $p \in \{0.01, 0.05, 0.1\}$, and (5) replicate (1–10). Because we used the genetic values directly in the GWAS, the LDSC slope yields an estimate of the genetic variance ($\hat{V}_g^{ldsc}$). For consistency, we also carried out GREML and HE regression estimation on the same data using GCTA (Yang, Lee, *et al.* 2011) (Supplementary Fig. 5).

To explore the inflation in GWAS test statistics observed under the CGF model (Fig. 6, Supplementary Fig. 3), we compared the GWAS effect sizes of all variants with their expected effects computed from their LD with the causal variants. Specifically, the expected effect of the jth SNP is $\mathbb{E}(\hat{\beta}_j) = \sum_{i=1}^{pM} \beta_i r_{ij}$ where $\beta_i$ is the true (simulated) effect of the ith causal variant and $r_{ij}$ is the genotypic correlation between them. We compared the expected and estimated GWAS effects for a single replicate of genotypes simulated under the HI and CGF models for generations 2 and 20 since admixture. These results are shown in Supplementary Fig. 4.

## Estimating variance explained by GWAS SNPs

We retrieved the summary statistics of 26 traits from GWAS catalog (Sollis *et al.* 2023). Full list of traits and the source papers (Hoffmann *et al.* 2018; Lona-Durazo *et al.* 2019; Pulit *et al.* 2019; Warrington *et al.* 2019; Chen *et al.* 2020; Ju and Mathieson 2020; Surendran *et al.* 2020; Graham *et al.* 2021; Yengo *et al.* 2022) are listed in Supplementary Table 1. To maximize the number of variants discovered, we chose summary statistics from studies that were conducted in both European and multiancestry samples and that reported the following information: effect allele, effect size, *p*-value, and genomic position. For height, we used the conditional and joint (COJO) effect sizes reported in Yengo *et al.* (2022). For birth weight, we downloaded the data from the Early Growth Genetics (EGG) consortium website (Warrington *et al.* 2019) since the version reported on the GWAS catalog is incomplete. For skin pigmentation, we chose summary statistics from the UKB (Bycroft *et al.* 2018) released by the Neale Lab (http://www.nealelab.is/uk-biobank) and processed by Ju and Mathieson (2020) to represent effect sizes estimated among individuals of European ancestry. We also selected summary statistics from Lona-Durazo *et al.* (2019) where effect sizes were meta-analyzed across four admixed cohorts. Lona-Durazo *et al.* provide summary statistics separately with and without conditioning on rs1426654 and rs35397—two large effect variants in *SLC24A5* and *SLC45A2*. We used the "conditioned" effect sizes and added in the effects of rs1426654 and rs35397 to estimate genetic variance.

We selected independent hits for each trait by pruning and thresholding with PLINK v1.90b6.21 (Chang *et al.* 2015) in two steps as in Ju and Mathieson (2020). We used the genotype data of GBR from the 1000 genome project (Auton *et al.* 2015) as the LD reference panel. We kept only SNPs (indels were removed) that passed the genome-wide significant threshold (*--clump-p1 5e-8*) with a pairwise LD cutoff of 0.05 (*--clump-r2 0.05*) and a physical distance threshold of 250 kb (*--clump-kb 250*) for clumping. Second, we applied a second round of clumping (*--clump-kb 100*) to remove SNPs within 100 kb.

When GWAS was carried out separately in different ancestry cohorts in the same study, we used inverse-variance weighting to meta-analyze effect sizes for variants that were genome-wide significant (*p*-value $<5 \times 10^{-8}$) in at least one cohort. This allowed us to maximize the discovery of variants such as the Duffy null allele that are absent among individuals of European ancestry but polymorphic in other populations (McManus *et al.* 2017).

We used allele frequencies from the 1000 Genomes CEU and YRI to represent the allele frequencies of GWAS SNPs in Europeans and Africans, respectively, making sure that the alleles reported in the summary statistics matched the alleles reported in the 1000 Genomes. We estimated the global ancestry of ASW individuals ($N = 74$) with CEU and YRI individuals from 1000 genome (phase 3) using ADMIXTURE 1.3.0 (Alexander *et al.* 2009) with $k = 2$ and used it to calculate the mean (proportion of African ancestry = 0.767) and variance (0.018) of global ancestry in ASW. With the effect sizes, allele frequencies, and the mean and variance in ancestry, we calculated the four components of genetic variance using Equation 1. The confidence intervals were calculated using a parametric bootstrap by randomly sampling the estimated effect of each variant 100 times from a standard normal distribution with a mean and standard deviation equal to the effect size and standard error reported by the original GWAS.

Initially, the multiancestry summary statistics for a few traits (NEU, WBC, MON, MCH, and BAS) yielded values >1 for the proportion of variance explained. This is likely because, despite LD pruning, some of the variants in the model are not independent and tag large effect variants under divergent selection such as the Duffy null allele, leading to an inflated contribution of LD. We checked this by calculating the pairwise contribution, i.e. $\beta_i\beta_j(f_i^A - f_i^B)(f_j^A - f_j^B)$, of all SNPs in the model and show long-range positive LD between variants on chromosome 1 for NEU, WBC, and MON, especially with the Duffy null allele (Supplementary Fig. 9a–c). A similar pattern was observed on chromosome 16 for MCH, confirming our suspicion. This also suggests that for certain traits, pruning and thresholding approaches are not guaranteed to yield independent hits. To get around this problem, we retained only one association with the lowest *p*-value, each from chromosome 1 (rs2814778 for NEU, WBC, and MON) and chromosome 16 (rs13331259 for MCH) (Supplementary Fig. 9d). For BAS, we observed that the variance explained was driven by a rare variant (rs188411703, MAF = 0.0024) of large effect ($\beta = -2.27$). We believe this effect estimate to be inflated and therefore, we removed it from our calculation.

As a sanity check to ensure transferability of GWAS effect sizes, we reestimated the effects using genotypic and phenotypic data from admixed individuals from the All of Us Research Program (AoU) (The All of Us Research Program Investigators 2019). To restrict the analysis to individuals with primarily mixed African and European ancestry, we focused on 61,461 individuals who self-identified as Black or African American, excluding those of Hispanic or Latino ethnicity. Of those 61,461 individuals, 48,587 individuals had trait and covariate data and were used as our final cohort. Phenotypes were matched from the GWAS catalog to AoU, with codes found in Supplementary Table 1 (The All of Us Research Program Investigators 2019; Sollis *et al.* 2023). Of the 26 traits, 21 were available in AoU, all except skin pigmentation, birth weight, pulse pressure, red blood cell count distribution width, and mean platelet volume. When necessary, we converted all observations to the most frequent unit of measurement for each trait (e.g. for height, the most common unit of measure was centimeters). We visually removed any outliers that were extreme or where the unit value was misspecified (Supplementary Table 1). The GWAS variants were extracted from vcf files called by the AoU from whole-genome sequencing (WGS) data. We reestimated the effect sizes for GWAS variants for the 21 traits using the *--glm* flag in PLINK (Chang *et al.* 2015) with age, age$^2$, sex assigned at birth, and the first 16 genetic PCs included as covariates. All phenotypes and covariates were standardized to unit variance. We show that effect sizes reestimated in AoU are highly correlated with the original GWAS effects with the exception of a few outliers (Supplementary Fig. 10).

## Data availability

We carried out all analyses with R version 4.2.3 (R Core Team 2023), PLINK v1.90b6.21 and PLINK 2.0 (Purcell *et al.* 2007; Chang *et al.* 2015), GCTA v1.94.1 (Yang, Lee, *et al.* 2011), LDSC v1.0.1 (Bulik-Sullivan *et al.* 2015), Admix-kit v0.1.1 (Hou *et al.* 2024), and Haptools v0.5.0 (Massarat *et al.* 2023). All code is freely available on https://github.com/zaidilab/admix_heritability.git. Summary statistics for each trait can be found in Supplementary Table 1. This study used data from the All of Us Research Program's Controlled Tier Dataset v7, available to authorized users on the Researcher Workbench (The All of Us Research Program Investigators 2019).

Supplemental material available at GENETICS online.

## Acknowledgments

We thank Iain Mathieson, Doc Edge, and reviewers for helpful comments on the manuscript. The content of this paper is solely the responsibility of the authors and does not necessarily represent the official views of the National Institutes of Health. The All of Us Research Program would not be possible without the partnership of its participants (The All of Us Research Program Investigators 2019).

## Funding

This study was funded by National Institute of General Medical Sciences awards R00GM137076 to A.A.Z. and T32GM132063 to N.K.

## Conflicts of interest

The author(s) declare no conflicts of interest.

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

# Appendix

## Variance in ancestry

We denote variance and covariance with $\mathbb{V}(.)$ and $\mathbb{C}(.)$ and used the expressions in Zaitlen *et al.* (2017) to generate the expected value for the variance in ancestry, i.e. $\mathbb{V}(\theta)$. This is straightforward for the HI model, where at time $t$ $\mathbb{V}(\theta_t) = \mathbb{V}(\theta_{t-1})\frac{(1+P_{t-1})}{2}$. $P_t = \text{Cor}(\theta_m, \theta_f)$ measures the strength of assortative mating, i.e. the correlation between the ancestry across mating pairs $(\theta_m, \theta_f)$ at time $t$. For simplicity, we assumed this to be constant in every generation, i.e. $P_t = P_{t-1} = P$ following Zaitlen *et al.* (2017). Since our notation slightly differs from Zaitlen *et al.* (2017), we re-derived the expression for $V(\theta_t)$ for the CGF model where population B receives a constant amount $q$ of gene flow from population A in every generation. Note, that $\mathbb{E}(\theta_t) = q + (1-q)\mathbb{E}(\theta_{t-1})$. Then,

$$
\begin{aligned}
\mathbb{V}(\theta_t) &= \mathbb{E}(\theta_t^2) - \mathbb{E}(\theta_t)^2 \\
&= q + (1-q)\mathbb{E}\left[\left(\frac{\theta_{t-1}^m + \theta_{t-1}^f}{2}\right)\left(\frac{\theta_{t-1}^m + \theta_{t-1}^f}{2}\right)\right] - \{q + (1-q)\mathbb{E}(\theta_{t-1})\}^2 \\
&= q + \frac{(1-q)}{4}\{2\mathbb{E}(\theta_{t-1}^2) + 2\mathbb{E}(\theta_{t-1}^m \theta_{t-1}^f)\} - \{q^2 + 2q(1-q)\mathbb{E}(\theta_{t-1}) \\
&\quad + (1-q)^2 \mathbb{E}(\theta_{t-1})^2\} \\
&= q + \frac{1-q}{2}\mathbb{E}(\theta_{t-1}^2) + \frac{1-q}{2}\mathbb{E}(\theta_{t-1}^m \theta_{t-1}^f) - q^2 - 2q(1-q)\mathbb{E}(\theta_{t-1}) \\
&\quad - (1-q)^2 \mathbb{E}(\theta_{t-1})^2 \\
&= q(1-q) + \frac{1-q}{2}\{\mathbb{V}(\theta_{t-1}) + \mathbb{E}(\theta_{t-1})^2\} + \frac{1-q}{2}\{\mathbb{C}(\theta_{t-1}^m, \theta_{t-1}^f) \\
&\quad + \mathbb{E}(\theta_{t-1})^2\} - 2q(1-q)\mathbb{E}(\theta_{t-1}) - \mathbb{E}(\theta_{t-1})^2 \\
&= q(1-q) + \frac{1-q}{2}\mathbb{V}(\theta_{t-1}) + \frac{1-q}{2}\mathbb{E}(\theta_{t-1})^2 + \frac{1-q}{2}P_{t-1}\mathbb{V}(\theta_{t-1}) \\
&\quad + \frac{1-q}{2}\mathbb{E}(\theta_{t-1})^2 - 2q(1-q)\mathbb{E}(\theta_{t-1}) - \mathbb{E}(\theta_{t-1})^2 \\
&= q(1-q) + \frac{1-q}{2}\mathbb{V}(\theta_{t-1})\{1 + P_{t-1}\} + (1-q)\mathbb{E}(\theta_{t-1})^2 \\
&\quad - 2q(1-q)\mathbb{E}(\theta_{t-1}) - (1-q)^2\mathbb{E}(\theta_{t-1})^2 \\
&= q(1-q)\mathbb{E}(\theta_{t-1})^2 + q(1-q)\{1 - 2\mathbb{E}(\theta_{t-1})\} + \frac{1-q}{2}\mathbb{V}(\theta_{t-1})\{1 + P_{t-1}\}
\end{aligned}
$$

## Genetic variance

Let $y = g + e$, where $y$ is the phenotypic value of an individual, $g$ is the genotypic value, and $e$ is random error. We assume additive effects such that $g = \sum_{i=1}^{m} \beta_i x_i$ where $\beta_i$ is the effect size of the ith biallelic locus and $x_i \in \{0, 1, 2\}$ is the number of copies of the trait-increasing allele. Then, the genetic variance $V_g$ is:

$$
V_g = \mathbb{V}\left(\sum_{i=1}^{m} \beta_i x_i\right) = \sum_{i=1}^{m} \beta_i^2 \mathbb{V}(x_i) + \sum_{j \neq i} \beta_i \beta_j \mathbb{C}(x_i, x_j).
$$

In the following sections, we decompose $\mathbb{V}(x_i)$ and $\mathbb{C}(x_i, x_j)$ further as functions of ancestry.

### $\mathbb{V}(x_i)$

We first derive $\mathbb{V}(x_i)$ as a function of ancestry ($\theta$) using the law of total variance:

$$
\mathbb{V}(x_i) = \mathbb{E}_\theta\{\mathbb{V}(x_i|\theta)\} + \mathbb{V}\{\mathbb{E}_\theta(x_i|\theta)\},
$$

where $\mathbb{E}_\theta$ represents the expectation taken over $\theta$.

$\mathbb{E}_\theta\{\mathbb{V}(x_i|\theta)\}$. We derive $\mathbb{V}(x_i|\theta)$ by further conditioning on the local ancestry at each locus.

$$\mathbb{V}(x_i|\theta) = \mathbb{E}_\gamma\{\mathbb{V}(x_i|\gamma, \theta)\} + \mathbb{V}\{\mathbb{E}_\gamma(x_i|\gamma, \theta)\},$$

where $\mathbb{E}_\gamma$ represents expectation taken over local ancestry. Since we are interested in the variance at a single locus, we will ignore the subscript $i$ and denote the frequency of the trait-increasing allele in populations A and B with $f^A$ and $f^B$, respectively.

$$
\begin{aligned}
\mathbb{E}_\gamma\{\mathbb{V}(x_i|\gamma, \theta)\} &= \mathbb{V}(x_i|\gamma=0, \theta)\mathbb{P}(\gamma=0|\theta) + \mathbb{V}(x_i|\gamma=1, \theta) \\
&\quad \mathbb{P}(\gamma=1|\theta) + \mathbb{V}(x_i|\gamma=2, \theta)\mathbb{P}(\gamma=2|\theta) \\
&= 2f^B(1-f^B)(1-\theta)^2 + \{f^A(1-f^A) + f^B(1-f^B)\}2\theta(1-\theta) \\
&\quad + 2f^A(1-f^A)\theta^2 \\
&= (2f^B - 2f^{B^2})(1-2\theta+\theta^2) + (f^A - f^{A^2} + f^B - f^{B^2})(2\theta - 2\theta^2) \\
&\quad + (2f^A - 2f^{A^2})\theta^2 \\
&= 2f^B - 2\theta f^B - 2f^{B^2} + 2\theta f^{B^2} + 2\theta f^A - 2\theta f^{A^2} \\
&= 2f^B(1-\theta) - 2f^{B^2}(1-\theta) + 2\theta f^A(1-f^A) \\
&= 2f^B(1-f^B)(1-\theta) + 2\theta f^A(1-f^A).
\end{aligned}
$$

To derive $\mathbb{V}\{\mathbb{E}_\gamma(x|\gamma, \theta)\}$, note that

$$
\begin{aligned}
\mathbb{E}_\gamma(x|\gamma, \theta) &= \mathbb{E}_\gamma\{\mathbb{E}(x|\theta)\} \\
&= \mathbb{E}(x|\gamma=0, \theta)\mathbb{P}(\gamma=0|\theta) + \mathbb{E}(x|\gamma=1, \theta)\mathbb{P}(\gamma=1|\theta) \\
&\quad + \mathbb{E}(x|\gamma=2, \theta)\mathbb{P}(\gamma=2|\theta) \\
&= 2\theta f^A + 2(1-\theta)f^B.
\end{aligned}
$$

And,

$$
\begin{aligned}
\mathbb{V}\{\mathbb{E}_\gamma(x|\gamma, \theta)\} &= \left[\mathbb{E}(x|\gamma=0, \theta) - \mathbb{E}(x|\theta)\right]^2\mathbb{P}(\gamma=0|\theta) \\
&\quad + \left[\mathbb{E}(x|\gamma=1, \theta) - \mathbb{E}(x|\theta)\right]^2\mathbb{P}(\gamma=1|\theta) \\
&\quad + \left[\mathbb{E}(x|\gamma=2, \theta) - \mathbb{E}(x|\theta)\right]^2\mathbb{P}(\gamma=2|\theta) \\
&= \theta^2\left[2f^A - \{2\theta f^A + 2(1-\theta)f^B\}\right]^2 \\
&\quad + 2\theta(1-\theta)\left[f^A + f^B - \{2\theta f^A + 2(1-\theta)f^B\}\right]^2 \\
&\quad + (1-\theta)^2\left[2f^B - \{2\theta f^A + 2(1-\theta)f^B\}\right]^2 \\
&= 2\theta(1-\theta)(f^A - f^B)^2.
\end{aligned}
$$

Putting this together,

$$
\begin{aligned}
\mathbb{E}_\theta\{\mathbb{V}(x_i|\theta)\} &= \mathbb{E}_\theta\{2f^B(1-f^B)(1-\theta) + 2\theta f^A(1-f^A) \\
&\quad + 2\theta(1-\theta)(f^A - f^B)^2\} \\
&= 2f^B(1-f^B)\{1 - \mathbb{E}_\theta(\theta)\} \\
&\quad + 2\mathbb{E}_\theta(\theta)f^A(1-f^A) + 2\mathbb{E}_\theta(\theta - \theta^2)(f^A - f^B)^2 \\
&= 2f^B(1-f^B)\{1 - \mathbb{E}_\theta(\theta)\} + 2\mathbb{E}_\theta(\theta)f^A(1-f^A) \\
&\quad + 2\{\mathbb{E}_\theta(\theta) - \mathbb{E}_\theta(\theta^2)\}(f^A - f^B)^2 \\
&= 2f^B(1-f^B)\{1 - \mathbb{E}_\theta(\theta)\} + 2\mathbb{E}_\theta(\theta)f^A(1-f^A) \\
&\quad + 2\{\mathbb{E}_\theta(\theta) - \mathbb{V}(\theta) - \mathbb{E}_\theta(\theta)^2\}(f^A - f^B)^2 \\
&= 2f^B(1-f^B)\{1 - \mathbb{E}_\theta(\theta)\} + 2\mathbb{E}_\theta(\theta)f^A(1-f^A) \\
&\quad + 2\mathbb{E}_\theta(\theta)(1 - \mathbb{E}_\theta(\theta))(f^A - f^B)^2 - 2\mathbb{V}(\theta)(f^A - f^B)^2
\end{aligned}
$$

$\mathbb{V}\{\mathbb{E}_\theta(x_i|\theta)\}$. Recall from the previous section that $\mathbb{E}_\theta(x_i|\theta) = 2\theta f^A + 2(1-\theta)f^B$. Then,

$$
\begin{aligned}
\mathbb{V}\{\mathbb{E}_\theta(x_i|\theta)\} &= \mathbb{V}\{2\theta f^A + 2(1-\theta)f^B\} \\
&= 4\mathbb{V}(\theta)f^{A^2} + 4\mathbb{V}(1-\theta)f^{B^2} + 2\mathbb{C}(2\theta f^A, 2(1-\theta f^B)) \\
&= 4\mathbb{V}(\theta)f^{A^2} + 4\mathbb{V}(1-\theta)f^{B^2} - 8f^A f^B \mathbb{V}(\theta) \\
&= 4\mathbb{V}(\theta)(f^A - f^B)^2.
\end{aligned}
$$

We are now ready to express $\mathbb{V}(x_i)$:

$$
\begin{aligned}
\mathbb{V}(x_i) &= 2f^B(1-f^B)\{1 - \mathbb{E}_\theta(\theta)\} + 2\mathbb{E}_\theta(\theta)f^A(1-f^A) \\
&\quad + 2\mathbb{E}_\theta(\theta)(1 - \mathbb{E}_\theta(\theta))(f^A - f^B)^2 \\
&\quad - 2\mathbb{V}(\theta)(f^A - f^B)^2 + 4\mathbb{V}(\theta)(f^A - f^B)^2 \\
&= 2\mathbb{E}_\theta(\theta)f_i^A(1-f_i^A) + 2\{1 - \mathbb{E}_\theta(\theta)\}f_i^B(1-f_i^B) \\
&\quad + 2\mathbb{E}_\theta(\theta)\{1 - \mathbb{E}_\theta(\theta)\}(f_i^A - f_i^B)^2 - 2\mathbb{V}(\theta)(f_i^A - f_i^B)^2.
\end{aligned}
$$

Note, that we can also express $V(x_i)$ as:

$$\mathbb{V}(x_i) = 2f_i(1-f_i) + 2\mathbb{V}(\theta)(f_i^A - f_i^B)^2,$$

where the second term is the contribution of population structure to the genetic variance at locus $i$.

## $\mathbb{C}(x_i, x_j)$

We can derive $\mathbb{C}(x_i, x_j)$ using the law of total covariance:

$$
\begin{aligned}
\mathbb{C}(x_i, x_j) &= \mathbb{E}_\theta\{\mathbb{C}(x_i, x_j|\theta)\} + \mathbb{C}\{\mathbb{E}_\theta(x_i|\theta), \mathbb{E}_\theta(x_j|\theta)\} \\
&= 0 + \mathbb{C}\{2f_i^A\theta + 2f_i^B(1-\theta), 2f_j^A\theta + 2f_j^B(1-\theta)\} \\
&= \mathbb{C}(2f_i^A\theta, 2f_j^A\theta) + \mathbb{C}(2f_i^A\theta, 2f_j^B(1-\theta) \\
&\quad + \mathbb{C}(2f_i^B(1-\theta), 2f_j^A\theta) + \mathbb{C}(2f_i^B(1-\theta), 2f_j^B(1-\theta)) \\
&= 4\mathbb{V}(\theta)(f_i^A - f_i^B)(f_j^A - f_j^B)
\end{aligned}
$$

$\mathbb{E}_\theta\{\mathbb{C}(x_i, x_j|\theta)\} = 0$ because we assume that the loci are unlinked and therefore, $x_i$ and $x_j$ are conditionally independent. Putting this all together, we get the genetic variance in admixed populations as presented in the main text:

$$
\begin{aligned}
V_g &= \sum_{i=1}^m \beta_i^2 \mathbb{V}(x_i) + \sum_{j\neq i}\beta_i\beta_j\mathbb{C}(x_i, x_j) \\
&= \sum_{i=1}^m \beta_i^2 2\mathbb{E}_\theta(\theta)f_i^A(1-f_i^A) + \sum_{i=1}^m \beta_i^2 2\{1 - \mathbb{E}_\theta(\theta)\}f_i^B(1-f_i^B) \\
&\quad + \sum_{i=1}^m \beta_i^2 2\mathbb{E}_\theta(\theta)\{1 - \mathbb{E}_\theta(\theta)\}(f_i^A - f_i^B)^2 \\
&\quad + \sum_{i=1}^m \beta_i^2 2\mathbb{V}(\theta)(f_i^A - f_i^B)^2] + \sum_{j\neq i}\beta_i\beta_j 4\mathbb{V}(\theta)(f_i^A - f_i^B)(f_j^A - f_j^B).
\end{aligned}
$$

The only difference being that in the main text we use $\mathbb{E}$ instead of $\mathbb{E}_\theta$ for simplicity. With two "unadmixed" source populations with

equal number of individuals, $\mathbb{E}(\theta) = 0.5$ and $\mathbb{V}(\theta) = \mathbb{E}(\theta)\{1 - \mathbb{E}(\theta)\} = 0.25$ and $V_g$ reduces to:

$$
\begin{aligned}
V_g = \mathbb{V}\left(\sum_{i=1}^{m} \beta_i x_i\right) &= \sum_{i=1}^{m} \beta_i^2 \mathbb{V}(x_i) + \sum_{j \neq i} \beta_i \beta_j \mathbb{C}(x_i, x_j) \\
&= \sum_{i=1}^{m} \beta_i^2 \left[ f_i^A(1 - f_i^A) + f_i^B(1 - f_i^B) \right] \\
&\quad + \sum_{i=1}^{m} \beta_i^2 (f_i^A - f_i^B)^2 \\
&\quad + \sum_{i \neq j} \beta_i \beta_j (f_i^A - f_i^B)(f_j^A - f_j^B).
\end{aligned}
$$

## Genetic variance after correction for individual ancestry

It can also be helpful to decompose $V_g$ into components of variance explained by and variance orthogonal to ancestry:

$$
V(g) = \underbrace{\mathop{\mathrm{V}}_{\theta}\{\mathbb{E}(g|\theta)\}}_{\substack{\text{variance along}\\\text{ancestry axis}}} + \underbrace{\mathop{\mathbb{E}}_{\theta}\{\mathbb{V}(g|\theta)\}}_{\substack{\text{variance orthogonal}\\\text{to ancestry axis}}}.
$$

We can express the residual variance as:

$$
\begin{aligned}
\mathbb{E}_\theta\{\mathbb{V}(g|\theta)\} &= \mathbb{E}_\theta\left\{\mathbb{V}\left(\sum_{i=1}^{M} \beta_i^2 x_i | \theta\right)\right\} \\
&= \mathbb{E}_\theta\left\{\sum_{i=1}^{M} \beta_i^2 \mathbb{V}(x_i|\theta)\right\} + \mathbb{E}_\theta\left\{\sum_{i \neq j} \beta_i \beta_j \mathbb{C}(x_i, x_j|\theta)\right\} \\
&= \sum_{i=1}^{M} \beta_i^2 \mathbb{E}_\theta\{\mathbb{V}(x_i|\theta)\} + 0 \\
&= 2\mathbb{E}_\theta(\theta) \sum_{i=1}^{M} \beta^2 f_i^A(1 - f_i^A) + 2\{1 - \mathbb{E}_\theta(\theta)\} \sum_{i=1}^{M} \beta^2 f_i^B(1 - f_i^B) \\
&\quad + 2\mathbb{E}_\theta(\theta) \sum_{i=1}^{M} \beta^2 \{1 - \mathbb{E}_\theta(\theta)\}(f_i^A - f_i^B)^2 - 2\mathbb{V}(\theta) \sum_{i=1}^{M} \beta^2 (f_i^A - f_i^B)^2.
\end{aligned}
$$

Note, that this represents the following components of $V_g$: (1.1) + (1.2)–(1.3).

## Haseman–Elston regression

The HE estimator of $V_g$ is based on the regression of products of (centered) phenotypes $y_k y_l$ for all pairs of individuals $k \neq l$ on the corresponding entries of the GRM ($\psi$), where $\psi_{kl} = \frac{\sum_{i=1}^{m} z_{ik} z_{il}}{m}$ and $z_{ik}$ is the centered and scaled genotype of individual $k$ for locus $i$. In this and the following sections, we denote the effect sizes corresponding to the scaled genotypes as $u_i$ and show the impact of alternative scaling schemes of the genotype on the HE estimator.

$$
\begin{aligned}
\hat{V}_g &= \frac{\mathbb{C}(y_k y_l, \psi_{kl})}{\mathbb{V}(\psi_{kl})} \\
&= \frac{\mathbb{E}_{kl}(y_k y_l \psi_{kl}) - \mathbb{E}_{kl}(y_k y_l)\mathbb{E}_{kl}(\psi_{kl})}{\mathbb{E}_{kl}(\psi_{kl}^2) - \mathbb{E}(\psi_{kl})^2} \\
&= \frac{\mathbb{E}_{kl}(y_k y_l \psi_{kl})}{\mathbb{E}_{kl}(\psi_{kl}^2)} = \frac{\mathbb{E}_{kl}\left(\sum_{i=1}^{m} u_i z_{ik} \sum_{i=1}^{m} u_i z_{il} \psi_{kl}\right)}{\mathbb{E}_{kl}(\psi_{kl}^2)} \\
&= \frac{\mathbb{E}_{kl}\left(\sum_{i=1}^{m} u_i^2 z_{ik} z_{il} \psi_{kl}\right)}{\mathbb{E}_{kl}(\psi_{kl}^2)} + \frac{\mathbb{E}_{kl}\left(\sum_{i=1}^{m} \sum_{j \neq i} u_i u_j z_{ik} z_{jl} \psi_{kl}\right)}{\mathbb{E}_{kl}(\psi_{kl}^2)},
\end{aligned}
$$

where $\mathbb{E}_{kl}$ represents the expectation over all $k \times l$ pairwise comparisons between individuals and the first and second terms represent the genic and LD components of the estimator.

### No directional LD

First, let us assume a genetic architecture where the effect sizes are random and there is no LD contribution, i.e. $\mathbb{E}_{ij}(u_i u_j) = 0$. We can further simplify the estimator as follows:

$$
\begin{aligned}
\hat{V}_g &= \frac{\mathbb{E}_{kl}\left(\sum_{i=1}^{m} u_i^2 z_{ik} z_{il} \psi_{kl}\right)}{\mathbb{E}_{kl}(\psi_{kl}^2)} + \frac{\mathbb{E}_{kl}\left(\sum_{i=1}^{m} \sum_{j \neq i} u_i u_j z_{ik} z_{jl} \psi_{kl}\right)}{\mathbb{E}_{kl}(\psi_{kl}^2)} \\
&= \frac{\mathbb{E}(u_i^2)\mathbb{E}_{kl}\left(\sum_{i=1}^{m} z_{ik} z_{il} \psi_{kl}\right)}{\mathbb{E}_{kl}(\psi_{kl}^2)} + \frac{\mathbb{E}(u_i u_j)\mathbb{E}_{kl}\left(\sum_{i=1}^{m} \sum_{j \neq i} z_{ik} z_{jl} \psi_{kl}\right)}{\mathbb{E}_{kl}(\psi_{kl}^2)} \\
&= \frac{\mathbb{E}_i(u_i^2)\mathbb{E}_{kl}(m\psi_{kl}^2)}{\mathbb{E}_{kl}(\psi_{kl}^2)} + 0 = m\mathbb{E}_{ij}(u_i^2).
\end{aligned}
$$

Thus, in the absence of directional LD, the estimator is a function of $\mathbb{E}(u_i^2)$.

*Scaling by $2f_i(1 - f_i)$.* With the standard scaling, the genotype at a given locus $i$ is $z_i = \frac{x_i - 2f_i}{\sqrt{2f_i(1 - f_i)}}$, where $f_i$ is the frequency of the allele in the population. Under the random-effects model, this is equivalent to saying that the unscaled effects are $\beta_i \sim \mathcal{N}(0, \frac{\sigma_u^2}{2mf_i(1 - f_i)})$ and the scaled effects are $u_i \sim \mathcal{N}(0, \frac{\sigma_u^2}{m})$. In a panmictic population, $V_g = \sum_{i=1}^{m} \beta_i^2 \mathbb{V}(x_i) = \sum_{i=1}^{m} \frac{\sigma_u^2}{2mf_i(1 - f_i)} 2f_i(1 - f_i) = \sigma_u^2$. Thus, in a panmictic population, $\mathbb{E}(\hat{V}_g) = \sigma_u^2 = V_g$, i.e. the standard scaling yields unbiased estimates of the genetic variance. In an admixed population,

$$
\begin{aligned}
V_g &= \sum_{i=1}^{m} \beta_i^2 \{2f_i(1 - f_i) + 2\mathbb{V}(\theta)(f_i^A - f_i^B)^2\} \\
&= \sum_{i=1}^{m} \frac{\sigma_u^2}{2mf_i(1 - f_i)} \{2f_i(1 - f_i) + 2\mathbb{V}(\theta)(f_i^A - f_i^B)^2\} \\
&= \frac{\sigma_u^2}{m} \sum_{i=1}^{m} \{1 + \mathbb{V}(\theta) \frac{(f_i^A - f_i^B)^2}{f_i(1 - f_i)}\} \\
&= \sigma_u^2 + \underbrace{\mathbb{V}(\theta) \frac{\sigma_u^2}{m} \sum_{i=1}^{m} \frac{(f_i^A - f_i^B)^2}{f_i(1 - f_i)}}_{\substack{\text{contribution of population}\\\text{structure to the genic variance}}}
\end{aligned}
$$

Thus, with the standard scaling, HE regression does not capture the contribution of population structure and therefore, gives a biased estimate of $V_g$.

*Scaling by $\mathbb{V}(x_i)$.* Next, we consider the case where the genotypes are standardized instead by the sample variance, i.e. $z_{kl} = \frac{x_{ik} - 2f_i}{\sqrt{\mathbb{V}(x_i)}}$ such that $\mathbb{V}(z_i) = 1$. We can derive $\mathbb{E}(u_i^2)$ corresponding to this scaling by noting that the genetic variance is invariant under linear transformations of the genotype (de los Campos *et al.* 2015):

$$
\begin{aligned}
\sum_{i=1}^{m} \beta_i^2 \mathbb{V}(x_i) &= \sum_{i=1}^{m} u_i^2 \mathbb{V}(z_i) \\
m\mathbb{E}(u_i^2) &= \sigma_u^2 + \mathbb{V}(\theta) \frac{\sigma_u^2}{m} \sum_{i=1}^{m} \frac{(f_i^A - f_i^B)^2}{f_i(1 - f_i)} \\
\mathbb{E}(u_i^2) &= \frac{\sigma_u^2}{m} + \mathbb{V}(\theta) \frac{\sigma_u^2}{m^2} \sum_{i=1}^{m} \frac{(f_i^A - f_i^B)^2}{f_i(1 - f_i)}.
\end{aligned}
$$

Then, the HE estimator becomes:

$$\hat{V}_g = m\mathbb{E}(u_i^2)$$
$$= m\left(\frac{\sigma_u^2}{m} + \mathbb{V}(\theta)\frac{\sigma_u^2}{m^2}\sum_{i=1}^m \frac{(f_i^A - f_i^B)^2}{f_i(1-f_i)}\right)$$
$$= \sigma_u^2 + \mathbb{V}(\theta)\frac{\sigma_u^2}{m}\sum_{i=1}^m \frac{(f_i^A - f_i^B)^2}{f_i(1-f_i)}.$$

Which provides an unbiased estimate of the genic variance. It is important to note that even though we assumed effect sizes under a random-effect model, the above result holds under a fixed-effect model as long as there is no directional LD. We discuss the implications of directional LD in the following section.

### Directional LD

Under the standard random-effect model, the effect sizes are assumed to be independent *in expectation*. We discussed in the main text how certain processes (e.g. selection and assortative mating) can induce directional LD across causal loci. But directional LD might arise even for neutral traits and under the random-effects model for any given realization of effects. This can lead to biases in both HE and GREML estimates of $V_g$, though the direction and reason for bias is different for the two methods. GREML does not have a closed-form solution so the exact estimand is difficult to derive. Here, we develop some intuition for HE regression.

*Scaling by $\mathbb{V}(x_i)$.* To do this, let $\boldsymbol{u}' = [u_1, u_2, \ldots, u_m]$ represent the vector of a given realization of (fixed) effects corresponding to the standardized genotypes such that each locus contributes equally to $\sigma_u^2$, the genic variance, i.e. $u_i^2 = \frac{\sigma_u^2}{m}$. Let there be positive LD across loci such that all cross-product terms are $u_i u_j = \frac{\sigma_u^2}{m}$. Then, the genetic variance explained by all loci is:

$$V_g = \sum_{i=1}^m u_i^2 \mathbb{V}(z_i) + \sum_{j \neq i} u_i u_j \mathbb{C}(z_i, z_j)$$
$$= \sum_{i=1}^m u_i^2 + \sum_{j \neq i} u_i u_j \mathbb{C}(z_i, z_j) \qquad \text{(A1)}$$
$$= \sigma_u^2 + \frac{\sigma_u^2}{m}\sum_{j \neq i}\mathbb{C}(z_i, z_j)$$

where $\mathbb{C}(z_i, z_j)$ is the LD between the $i$th and $j$th loci that ranges from 0 (no LD) to 1 (perfect LD). Thus, the LD contribution to $V_g$ ranges from 0 to $(m-1)\sigma_m^2$. In comparison, the HE estimator is:

$$\hat{V}_g = \frac{\mathbb{E}_{kl}\left(\sum_{i=1}^m u_i^2 z_{ik} z_{il} \psi_{kl}\right)}{\mathbb{E}_{kl}(\psi_{kl}^2)} + \frac{\mathbb{E}_{kl}\left(\sum_{i=1}^m \sum_{j\neq i} u_i u_j z_{ik} z_{jl} \psi_{kl}\right)}{\mathbb{E}_{kl}(\psi_{kl}^2)}$$
$$= \frac{\mathbb{E}_{kl}\left(\sum_{i=1}^m \frac{\sigma_u^2}{m} z_{ik} z_{il} \sum_{w=1}^m z_{wk} z_{wl}/m\right)}{\mathbb{E}_{kl}\left(\sum_{i=1}^m z_{ik} z_{il}/m \sum_{w=1}^m z_{wk} z_{wl}/m\right)}$$
$$\qquad\qquad\qquad\qquad\qquad\qquad\qquad\qquad \text{(A2)}$$
$$+ \frac{\mathbb{E}_{kl}\left(\sum_{i=1}^m \sum_{j\neq i} \frac{\sigma_u^2}{m} z_{ik} z_{jl} \sum_{w=1}^m z_{wk} z_{wl}/m\right)}{\mathbb{E}_{kl}\left(\sum_{i=1}^m z_{ik} z_{il}/m \sum_{w=1}^m z_{wk} z_{wl}/m\right)}$$
$$= \sigma_u^2 + \sigma_u^2 \frac{\mathbb{E}_{kl}\left(\sum_{i=1}^m \sum_{j\neq i} z_{ik} z_{jl} z_{wk} z_{wl}\right)}{\mathbb{E}_{kl}\left(\sum_{i=1}^m z_{ik} z_{il} \sum_{w=1}^m z_{wk} z_{wl}\right)}.$$

This shows that the bias due to directional LD in the HE estimate of $V_g$ does not come from the genic, but the LD component. When there is no LD, e.g. if the population has reached equilibrium after generations of random mating, this component goes to zero and both the estimate and $V_g$ converge to the same value—the genic

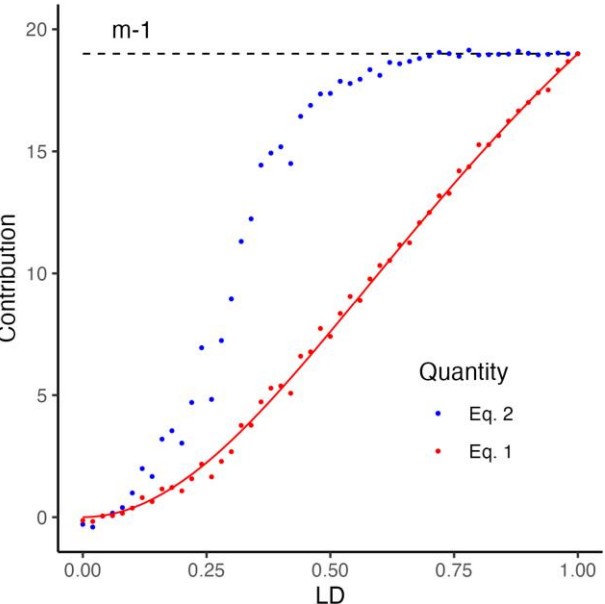

**Fig. A1.** The behavior of the LD contribution (y-axis) to the genetic variance (red) and the HE regression estimate (blue) as a function of LD, i.e. $\mathbb{C}(z_i, z_j)$ (x-axis). Each point represents the contribution calculated from a random draw of genotypes, given $\mathbb{C}(z_i, z_j) \propto (f_i^A - f_i^B)(f_j^A - f_j^B)$. The red line represents the expected LD contribution, and the black dashed line represents the contribution expected in the case of perfect LD.

variance. The LD component is maximum when the $i$th and $j$th loci are in perfect LD. In this case, $i$ and $j$ are exchangeable and the second term of the estimator reduces to $(m-1)\sigma_m^2$. Thus, HE regression should give an unbiased estimate of $V_g$, even in the presence of directional LD, but only when LD is perfect. For any other value $0 < C(z_i, z_j) < 1$, the estimate is biased (Fig. A1). An interpretable, analytical derivation of the second term in Equation A2 is complicated but we illustrate the bias with simulations below. For unlinked markers, $\mathbb{C}(z_i, z_j)$ is a function of $4\mathbb{V}(\theta)(f_i^A - f_i^B)(f_j^A - f_j^B)$ (see $\mathbb{V}(x_i)$). Perfect LD arises when (1) both $f_i^A - f_i^B = 1$ and $f_j^A - f_j^B = 1$ and (2) $\mathbb{V}(\theta)$ is maximum, which occurs at the time of admixture when source populations mix equally, i.e. $\mathbb{E}(\theta) = 0.5$. To generate a range of LD, we simulated an admixed population ($N = 1{,}000$) with equal number of individuals from populations A and B. Thus, $4\mathbb{V}(\theta) = 4\mathbb{E}(\theta)\{1 - \mathbb{E}(\theta)\} = 1$. We simulated genotypes for each individual at 50 "causal" loci where the difference between the frequencies in the source populations, $f_i^A - f_i^B \in [0, 1]$ with the condition that $\frac{f_i^A + f_i^B}{2} = 0.5$. We assigned each locus the same effect size (on the variance-standardized scale) of $+1/\sqrt{m}$ summing up to a genic variance of 1. The positive sign ensures positive LD across loci, i.e. all off-diagonal elements of $\boldsymbol{uu}'$ are set to $1/m$. For each simulation, we computed the expected and estimated LD component using the second terms in Equations A1 and A2, respectively, and averaged the results over 100 replications.

*Scaling by LD.* In the main text, we showed that standardizing the genotypes at a locus by its covariance with other loci accounts for the bias for GREML and HE estimators. More specifically, the "LD-scaled" genotypes can be written as $\boldsymbol{Z} = (\boldsymbol{X} - 2\boldsymbol{P})\boldsymbol{U}^{-1}$ where $\boldsymbol{P}$ is an $n \times m$ matrix such that all elements of the $i$th column contain the frequency of the $i$th SNP and $\boldsymbol{U}$ is the (upper triangular) square root of the LD matrix, i.e. $\boldsymbol{\Sigma} = \boldsymbol{U}'\boldsymbol{U}$. Under this scheme, the standardized genotypes are uncorrelated and therefore, the second term in Equations A1 and A2 are zero. This reduces the estimator to the

first term, representing the sum of squares of effect sizes, i.e. $u'u = \sum_{i=1}^{m} u_i^2$. The effect sizes corresponding to the LD scaled genotypes are $u = U\beta$ and the sum of squares is:

$$u'u = (U\beta)'(U\beta) = \beta'U'U\beta = \beta\Sigma\beta = \sum_{i=1}^{m}\sum_{j=1}^{m}\beta_i\beta_j\mathbb{C}(x_i, x_j)$$

Which captures both the genic and LD contributions and therefore, provides an unbiased estimate of $V_g$.

### Genetic variance after correction for individual ancestry

In GREML it is common to include ancestry or principal components of the GRM to correct for population structure, which can lead to inflated estimates of heritability (Browning SR and Browning 2011; Yang, Manolio, *et al.* 2011). In HE regression, a moment-matching approach (MMHE) can be used to account for covariates (Ge *et al.* 2017; Lin M *et al.* 2021). To explain this, we use matrix notation. Recall the generative model $y = g + e = Zu + e$ where $y$ is an $n \times 1$ (centered) phenotype vector, $Z$ is $n \times m$ matrix of standardized genotypes, and $u$ is an $m \times 1$ vector of effect sizes. In the absence of covariates, HE regression estimates $\hat{\sigma}_u^2$ and $\hat{\sigma}_e^2$ by regressing the empirical covariance of the phenotypes onto the GRM: $\text{vec}(yy') = \text{vec}(Zuu'Z') + \text{vec}(ee')$ where $\text{vec}(.)$ is the vectorization operator. If the effects are independent, this simplifies to $\sigma_u^2\text{vec}(\psi) + \sigma_e^2\text{vec}(I)$. Then $\hat{\sigma}_u^2$ and $\hat{\sigma}_e^2$ can be obtained from ordinary least squares (OLS).

Conceptually, the MMHE approach adjusts for covariates (e.g. sex, age, and ancestry) by projecting them out of both the phenotypes and genotypes (Ge *et al.* 2017). We denote the projection matrix as $P = C(C'C)^{-1}C'$ where $C$ is the $n \times q$ design matrix of covariates. In our case, $C = [1 \, \theta]$ where $\theta$ is the $n \times 1$ vector of individual ancestry. Then, let $\dot{y} = My$ and $\dot{Z} = MZ$ can be thought of as the residuals of a regression of $y$ and $Z$, respectively, on $\theta$ and $\dot{\psi} = \dot{Z}\dot{Z}'$ as the corresponding kinship matrix. Then, the ancestry-adjusted estimate of $\sigma_u^2$ can be obtained by regressing $\text{vec}(\dot{y}\dot{y}')$ on $\text{vec}(\dot{\psi}')$:

$$
\begin{aligned}
\text{vec}(\dot{y}\dot{y}') &= \text{vec}(\dot{g}\dot{g}') + \text{vec}(\dot{e}\dot{e}') \\
&= \text{vec}((MZu)(MZu)') + \sigma_e^2 I \\
&= \text{vec}(MZuu'Z'M) + \sigma_e^2 I \\
&= \text{vec}(MZ\sigma_u^2 Z'M) + \sigma_e^2 I \\
&= \text{vec}(\dot{Z}\sigma_u^2\dot{Z}') + \sigma_e^2 I \\
&= \sigma_u^2\text{vec}(\dot{\psi}) + \sigma_e^2 I_{(n-q)}.
\end{aligned}
$$

The simplification in line 3 follows if we assume that (1) the off-diagonal terms of $uu'$ are 0, i.e. the effects are independent, and (2) all loci contribute equally to the genetic variance. The OLS estimates of $\sigma_u^2$ and $\sigma_e^2$ can be obtained by solving the following linear system (Ge *et al.* 2017; Lin Z *et al.* 2022):

$$
\begin{bmatrix} \text{tr}(M\psi M\psi) & \text{tr}(M\psi) \\ \text{tr}(\psi) & n-q \end{bmatrix}\begin{bmatrix} \sigma_u^2 \\ \sigma_e^2 \end{bmatrix}\begin{bmatrix} y'M\psi My \\ y'My, \end{bmatrix}
$$

where $\text{tr}(.)$ represents the trace of the matrix. In practice, we use the more efficient algorithm in Ge *et al.* (2017), which we used to estimate $\hat{\sigma}_u^2$.

### LD score regression

LD score regression (LDSC) estimates $h_{\text{snp}}^2$ under a random-effects model from GWAS summary statistics using method-of-moments. Here, we provide intuition for how LDSC can be biased in the presence

of directional LD. We define $\chi_k^2$ as the marginal association statistic of the $k$th marker and $l_k = \sum_{w=1}^{W} r_{wk}^2$ as its LD score. Thus, LDSC estimates $h_{\text{snp}}^2$ from the regression slope of $\chi_k^2$ on $l_k$ over all $k$ markers. To understand this, let us denote the estimated marginal effect of the $k$th marker as a function of its true marginal effect and some estimation error: $\hat{\tau}_k = \tau_k + \varepsilon_k$ assuming for simplicity that there is no residual bias due to stratification in the upstream GWAS. Then, $\mathbb{E}(\hat{\tau}_k) = \tau_k$ and $\mathbb{E}(\hat{\tau}_k^2) = \tau_k^2 + \mathbb{E}(\varepsilon^2) = \tau_k^2 + SE^2(\hat{\tau}_k)$. Furthermore, $\tau_k = \sum_{i=1}^{m} r_{ik}u_i$, where $r_{ik}$ is the LD between the $k$th marker and the $i$th causal locus. If the individual causal effects are small, then $SE^2(\hat{\tau}_k) \approx 1/n$ and

$$\mathbb{E}(\chi_k^2) = \frac{\mathbb{E}(\hat{\tau}_k^2)}{SE^2(\hat{\tau}_k)} = \frac{\tau_k^2 + SE^2(\hat{\tau}_k)}{SE^2(\hat{\tau}_k)} = \frac{\tau_k^2 + 1/n}{1/n} = n\tau_k^2 + 1.$$

The expected slope of LD Score regression is:

$$
\begin{aligned}
\hat{\beta}_{\chi^2,l} &= \frac{\mathbb{C}(\chi_k^2, l_k)}{\mathbb{V}(l_k)} = \frac{\mathbb{C}(n\tau_k^2 + 1, l_k)}{\mathbb{V}(l_k)} = \frac{\mathbb{C}\left(\left(\sum_{i=1}^{m} r_{ik}u_i\right)^2, l_k\right)}{\mathbb{V}(l_k)} \\
&= \frac{n\mathbb{C}\left(\sum_{i=1}^{m} r_{ik}^2 u_i^2 + 2\sum_{i=1}^{m}\sum_{j<i} r_{ik}r_{jk}u_iu_j, \ \sum_{i=1}^{m} r_{ik}^2\right)}{\mathbb{V}(l_k)} \\
&= \frac{n\mathbb{C}\left(\sum_{i=1}^{m} r_{ik}^2 u_i^2, \ \sum_{i=1}^{m} r_{ik}^2\right) + n\mathbb{C}\left(2\sum_{i=1}^{m}\sum_{j<i} r_{ik}r_{jk}u_iu_j, \ \sum_{i=1}^{m} r_{ik}^2\right)}{\mathbb{V}(l_k)}.
\end{aligned}
$$

As with GREML and HE regression, LDSC assumes a polygenic, random-effects model where $u_i \sim \mathcal{N}(0, \frac{\sigma_u^2}{m})$. As such, the cross-product term inside the brackets is zero over variants $i$ and $j$, and the slope reduces to:

$$\hat{\beta}_{\chi^2,l} = \frac{n\mathbb{C}\left(\sum_{i=1}^{m} r_{ik}^2 u_i^2, \ \sum_{i=1}^{m} r_{ik}^2\right)}{\mathbb{V}(l_k)} = \frac{n\sigma_u^2}{m}$$

from which $\hat{V}_g$ and therefore, $\hat{h}_{\text{snp}}^2$ can be derived. To understand the behavior of LDSC under directional LD, consider a simple generative model with $m$ causal variants, each with the same scaled (fixed) effect $u$ in positive LD such that $V_g = \sum_{i=1}^{m} u_i^2 \mathbb{V}(z_i) + 2\sum_{i=1}^{m}\sum_{j<i}\mathbb{C}(z_i, z_j) = \sum_{i=1}^{m} u_i^2 + 2\sum_{i=1}^{m}\sum_{j<i} u_iu_jr_{ij} = mu^2 + 2u^2\sum_{i=1}^{m}\sum_{j<i} r_{ij}$. We further assume that causal variants are unlinked, that any LD between them is due to population structure, and that each causal variant is tagged by $W$ markers. The marginal effect of a marker is a function of its LD with the causal loci, i.e. $\tau_k = \sum_{i=1}^{m} r_{ik}u_i = u\sum_{i=1}^{m} r_{ik}$. However, if, as per standard practice, individual ancestry was included as a covariate in the GWAS, then the marginal effect of the marker will only absorb the effect of the causal variant nearby but not of variants in LD due to population structure. Thus, if we note $r_{kc}$ as the LD between the $k$th marker and the causal variant in its vicinity, $\tau_k = ur_{kc}$, $\tau_k^2 = r_{kc}^2 u^2$, and $\mathbb{E}(\chi_k^2) = nr_{kc}^2 u^2 + 1$. In practice, LDSC regresses $\chi_k^2$ on $l_k = \sum_{w=1}^{W} r_{kw}^2$:

$$
\begin{aligned}
\text{LDSC} &= \frac{\mathbb{C}(\chi_k^2, l_k)}{\mathbb{V}(l_k)} = \frac{\mathbb{C}(\chi_k^2, \sum_w^W r_{kw}^2)}{\mathbb{V}(l_k)} = \frac{\sum_w^W \mathbb{C}(\chi_k^2, r_{kw}^2)}{\mathbb{V}(l_k)} \\
&= \frac{\mathbb{C}(\chi_k^2, r_{kc}^2)}{\mathbb{V}(l_k)} + \frac{\sum_{w\neq c}\mathbb{C}(\chi_k^2, r_{kw}^2)}{\mathbb{V}(l_k)} = \frac{nu^2 r_{kc}^2}{\mathbb{V}(l_k)}
\end{aligned}
$$

### Effect size of local ancestry

We define local ancestry $\gamma_i \in \{0, 1, 2\}$ as the number of alleles at locus $i$ that trace their ancestry to population A. Thus, the local

ancestry at locus $i$ in individual $k$ is a Binomial random variable with $\mathbb{E}(\gamma_{i,k}) = 2\theta_k$. We define the ancestry value of an individual as the weighted sum of their local ancestry: $\sum_{i=1}^{m} \phi_i \gamma_i$ where $\phi_i = \beta_i(f_i^B - f_i^A)$.

To show this, note that $\phi = \mathbb{E}(y|\gamma = 1) - \mathbb{E}(y|\gamma = 0)$ where $\mathbb{E}(y|\gamma = 1) = \int_{-\infty}^{\infty} y h(y|\gamma = 1)$ and $h$ is a density function. Our goal is to express $\phi$ in terms of $\beta$, which is equal to $\mathbb{E}(y|x = 1) - \mathbb{E}(y|x = 0)$. Furthermore, $\mathbb{E}(y|x = 1) = \int_{-\infty}^{\infty} y h(y|x = 1)$. We can express $h(y|\gamma)$ in terms of $h(y|x)$ as follows:

$$
\begin{aligned}
h(y|\gamma = 1) &= h(y|x = 0)\mathbb{P}(x = 0|\gamma = 1) + h(y|x = 1)\mathbb{P}(x = 1|\gamma = 1) \\
&\quad + h(y|x = 2)\mathbb{P}(x = 2|\gamma = 1) \\
&= h(y|x = 0)2(1 - f^A)(1 - f^A) \\
&\quad + h(y|x = 1)\{f^A(1 - f^B) \\
&\quad + f^B(1 - f^A)\} + h(y|x = 2)2f^A f^B \\
\mathbb{E}(y|\gamma = 1) &= \int_{-\infty}^{\infty} y h(y|\gamma = 1)dy \\
&= (1 - f^A)(1 - f^B)\int_{-\infty}^{\infty} y h(y|x = 0)dy \\
&\quad + \{f^A(1 - f^B) + f^B(1 - f^A)\}\int_{-\infty}^{\infty} y h(y|x = 1)dy \\
&\quad + f^A f^B \int_{-\infty}^{\infty} y h(y|x = 2)dy \\
&= (1 - f^A)(1 - f^B)\mathbb{E}(y|x = 0) + \{f^A(1 - f^B) \\
&\quad + f^B(1 - f^A)\}\mathbb{E}(y|x = 1) + f^A f^B \mathbb{E}(y|x = 2) \\
&= 0 + \{f^A(1 - f^B) + f^B(1 - f^A)\}\beta + f^A f^B 2\beta \\
&= \beta f^A + \beta f^B
\end{aligned}
$$

Similary, $\mathbb{E}(y|\gamma = 0) = 2\beta f^B$ and $\phi = \mathbb{E}(y|\gamma = 1) - \mathbb{E}(y|\gamma = 0) = \beta(f^B - f^A)$

## Genetic variance due to local ancestry

$$
\begin{aligned}
V_\gamma &= \mathbb{V}\left(\sum_{i=1}^{m} \phi_i \gamma_i\right) \\
&= \sum_{i=1}^{m} \phi_i^2 \mathbb{V}(\gamma_i) + \sum_{i=1}^{m}\sum_{j\neq i} \phi_i \phi_j \mathbb{C}(\gamma_i, \gamma_j)
\end{aligned}
\tag{A3}
$$

We use the law of total variance and covariance to derive $\mathbb{V}(\gamma_i)$ and $\mathbb{C}(\gamma_i, \gamma_j)$:

$$
\begin{aligned}
\mathbb{V}(\gamma_i) &= \mathbb{E}\{\mathbb{V}(\gamma_i|\theta)\} + \mathbb{V}\{\mathbb{E}(\gamma_i|\theta)\} \\
&= \mathbb{E}\{2\theta(1 - \theta)\} + \mathbb{V}(2\theta) \\
&= 2\mathbb{E}(\theta) - 2\mathbb{E}(\theta^2) + 4\mathbb{V}(\theta) \\
&= 2\mathbb{E}(\theta) - 2\mathbb{V}(\theta) - 2\mathbb{E}(\theta)^2 + 4\mathbb{V}(\theta) \\
&= 2\mathbb{E}(\theta)\{1 - \mathbb{E}(\theta)\} + 2\mathbb{V}(\theta) \\
\mathbb{C}(\gamma_i, \gamma_j) &= \mathbb{E}\{\mathbb{C}(\gamma_i, \gamma_j|\theta)\} + \mathbb{C}\{\mathbb{E}(\gamma_i, \gamma_j|\theta)\} \\
&= 0 + \mathbb{C}(2\theta, 2\theta) = 4\mathbb{V}(\theta) \\
V_\gamma &= 2\mathbb{E}(\theta)\{1 - \mathbb{E}(\theta)\}\sum_{i=1}^{m} \phi_i^2 + 2\mathbb{V}(\theta)\sum_{i=1}^{m} \phi_i^2 + 4\mathbb{V}(\theta)\sum_{i=1}^{m}\sum_{j\neq i} \phi_i \phi_j \\
&= 2\mathbb{E}(\theta)\{1 - \mathbb{E}(\theta)\}\sum_{i=1}^{m} \beta_i^2 (f_i^B - f_i^A)^2 \\
&\quad + 2\mathbb{V}(\theta)\sum_{i=1}^{m} \beta_i^2 (f_i^B - f_i^A)^2 \\
&\quad + 4\mathbb{V}(\theta)\sum_{i=1}^{m}\sum_{j\neq i} \beta_i \beta_j (f_i^B - f_i^A)(f_j^B - f_j^A).
\end{aligned}
$$

*Editor: Y. Li*