## [Peer Review File · Genetics]

Interpreting SNP heritability in admixed populations

Jinguo Huang, Nicole Kleman, Saonli Basu, Mark Shriver, and Arslan Zaidi

NOTE: The reviews and decision letters are unedited and appear as submitted by the reviewers.

In extremely rare instances and as determined by a Senior Editor or the EIC, portions of a review may be redacted. If a review is signed, the reviewer has agreed to no longer remain anonymous.

The review history appears in chronological order.

Review Timeline:

Submission Date:	2024-08-02
Editorial Decision:	2024-09-19
Resubmission Received:	2025-03-31
Accepted:	2025-04-29

Editorial Note: Reviews of this manuscript were transferred to GENETICS from another journal. Decision letters and reviews from outside of GENETICS have been redacted from the Peer Review History document to remove material for which no permission to publish was obtained.

September 19, 2024

GENETICS-2024-307341
Interpreting SNP heritability in admixed populations

Dear Dr. Zaidi:

Two experts in the field have reviewed your manuscript, and I have read it as well. While your manuscript is not currently acceptable for publication in GENETICS, we would welcome a substantially revised manuscript.

Two experts in the field have reviewed your manuscript, and I have read it as well. I appreciate the rigor of your approach and the significance of the findings. I view it highly valuable for geneticists to realize the potential bias when estimating SNP heritability in admixed populations. While your manuscript is not currently acceptable for publication in GENETICS, we would welcome a substantially revised manuscript. Both reviewers have comments and concerns to be addressed in a revised manuscript. You can read their reviews at the end of this email. In particular, please carefully address the comments regarding more binary outcomes in addition to skin pigmentation, and assess the bias when LDSC and GENESIS are used. We look forward to receiving your revised manuscript. Please let the editorial office know approximately how long you expect to need for revisions.

Upon resubmission, please include:

1. A clean version of your manuscript;
2. A marked version of your manuscript in which you highlight significant revisions carried out in response to the major points raised by the editor/reviewers (track changes is acceptable if preferred);
3. A detailed response to the editor's/reviewers' feedback and to the concerns listed above. Please reference line numbers in this response to aid the editor and reviewers.

Your paper will likely be sent back out for review.

Additionally, please ensure that your resubmission is formatted for GENETICS
<https://academic.oup.com/genetics/pages/general-instructions>

Follow this link to submit the revised manuscript: Link Not Available

Sincerely,

Yun Li
Associate Editor
GENETICS

Approved by:
Hongyu Zhao
Senior Editor
GENETICS

Reviewer #1 :

In this manuscript, Huang et al. examined the bias of SNP heritability estimate in random-effect models in admixed populations. They showed with theoretical and simulation results that the widely-used GCTA-GREML estimator is biased due to LD contribution considering the evolutionary trait selection between the source populations. They also estimated the LD contribution of genome-wide significant SNPs for 26 quantitative traits from GWAS Catalog using 1000G ASW population. The results showed mostly little but large bias for skin pigmentation.

I have also carefully read the comments from previous reviewers and the authors' responses. They have done a decent job addressing these concerns. I would additionally suggest the authors trying some real GWAS analysis for binary diseases that show different prevalence between European and African populations to see if the LD contribution will be similarly obvious as for skin pigmentation.

Reviewer #2 :

Overall Thoughts:

The authors argue that heritability is biasedly estimated when using the genome-wide restricted maximum likelihood (GREML) procedure or the Haseman-Elston (HE) regression procedure in the case of admixed populations. To prove this, authors provide a general formula for the additive genetic variance of a quantitative trait in the case of a population coming from a mixture of two previously isolated populations and perform extensive simulations under this scenario to discuss the biased estimation of GREML and HE regression. But as this article serves to showcase a problem that exists and not to propose a method to solve this problem, the analysis of only two methods, one being implemented in only custom scripts, in my opinion, limits the scope of this manuscript. The derivations and implementation of the simulation studies appear largely correct, and my only comments are related to the scope of the manuscript as well as some small technical details.

Comment 1:

My first comment is with regards to the motivation of this work. There is large motivation in understanding how to analyze admixed populations as the increased globalization should also increase the numbers of individuals who are admixed. But currently the large diverse multi-ethnic cohorts in my experience have analyzed individuals from each ancestry separately and leveraged information from each model within ancestries to increase, for example, prediction performance of polygenic risk scores¹. Typically, methods avoid the analysis of admixed populations as defining them accurately is difficult as well as a lack of methodological development of methods.

Comment 2:

Throughout the manuscript there is a few notations that are not clearly defined when first introduced. Both m and F_{ST} are defined when they are introduced in equation 1 and the first paragraph of the results, respectively.

Comment 3:

Under Equation 1, the article examines the bias of two procedure GREML and HE regression in a variety of simulations and discusses where these methods may be biased. As this article goal is to "study the behavior of $[[h^2]]_{snp}$ in admixed populations", the analysis of only two methods greatly diminishes the scope of the article for users of other software. For example, LDSC uses a method of moments estimator and methods such as GENESIS use a Bayesian spike and slab prior to estimate heritability^{2,3}. From this article, a user of these methods has a very limited understanding of how methods they might be most comfortable are impacted. Further comparisons to a broader set of methods in either the results or discussion section will provide greatly increase the scope to the average reader.

Comment 4:

The article provides a clear conclusion the biases in estimation of heritability of the SNP occur when either using HE regression or GREML under differing true genetic architecture of the admixed population. The only time when bias was near 0, was in the case of scaling the genotypes by the LD matrix, which as properly noted is almost always impossible. Therefore, for a practitioner, there is an overall lack of general conclusion to garner from this manuscript. In a previous response to reviewers, the authors expressed that they still prefer GREML over HE regression, if this is the case, then there should be more structure in the discussion section about this to provide a clearer take away message for the reader. As the current discussion, provides confusion and the message of avoiding analyses of admixed populations.

References:

1. Zhang, H. et al. A new method for multi-ancestry polygenic prediction improves performance across diverse populations. *Nat Genet* 55, 1757-1768 (2023).
2. Zhang, Y., Qi, G., Park, J. H. & Chatterjee, N. Estimation of complex effect-size distributions using summary-level statistics from genome-wide association studies across 32 complex traits. *Nature Genetics* 2018 50:9 50, 1318-1326 (2018).
3. Bulik-Sullivan, B. et al. LD Score regression distinguishes confounding from polygenicity in genome-wide association studies. *Nat Genet* 47, 291-295 (2015).

We thank the reviewers for their time and feedback and for the editor's comments on the rigor of our work. We have taken the reviewers' suggestions into consideration and have made changes to the manuscript that have further improved its scope. A major update to the paper is the addition of a section analyzing the behavior of LD score regression in admixed populations. We have also made the following updates to further improve rigor and clarity:

- Updated the figure showing partitioning of variance components in GWAS SNPs to include 95% confidence intervals (now Fig. 8).
- Added Fig. S6, S7, S8 (supporting the LDSC analyses).
- Added Fig. S10 showing the correlation between previous GWAS effect sizes and effect sizes of the same SNPs re-estimated in the African Americans from the All of Us dataset as a sanity check to show portability of genetic architecture.
- Used fixed-effects throughout in simulating genetic architecture to give us more control over the simulations.

Our results and conclusions remain unchanged. We address the reviewers' comments in detail below and provide a .pdf file (compiled using latexdiff) to indicate where changes have been made.

Reviewer #1 :

In this manuscript, Huang et al. examined the bias of SNP heritability estimate in random-effect models in admixed populations. They showed with theoretical and simulation results that the widely-used GCTA-GREML estimator is biased due to LD contribution considering the evolutionary trait selection between the source populations. They also estimated the LD contribution of genome-wide significant SNPs for 26 quantitative traits from GWAS Catalog using 1000G ASW population. The results showed mostly little but large bias for skin pigmentation.

I have also carefully read the comments from previous reviewers and the authors' responses. They have done a decent job addressing these concerns. I would additionally suggest the authors trying some real GWAS analysis for binary diseases that show different prevalence between European and African populations to see if the LD contribution will be similarly obvious as for skin pigmentation.

We did not analyze binary traits in this paper because our generative model assumes a quantitative trait and there are complications related to scale and disease prevalence that make the calculation and interpretation of heritability for binary traits more complicated. We recognize that the extent to which directional LD contributes to differences in prevalence of binary traits between populations is an important question. But this is a non-trivial undertaking that will be the focus of a future manuscript.

Reviewer #2 :

Overall Thoughts:

The authors argue that heritability is biasedly estimated when using the genome-wide restricted maximum likelihood (GREML) procedure or the Haseman-Elston (HE) regression procedure in the case of admixed populations. To prove this, authors provide a general formula for the additive genetic variance of a quantitative trait in the case of a population coming from a mixture of two previously isolated populations and perform extensive simulations under this scenario to discuss the biased estimation of GREML and HE regression. But as this article serves to showcase a problem that exists and not to propose a method to solve this problem, the analysis of only two methods, one being implemented in only custom scripts, in my opinion, limits the scope of this manuscript. The derivations and implementation of the simulation studies appear largely correct, and my only comments are related to the scope of the manuscript as well as some small technical details.

We thank the reviewer for their careful reading and comments and address their comments pointwise below. In the revised manuscript, we have also used a published method (MMHE)¹ instead of custom scripts to implement the ancestry-adjusted Haseman-Elston regression.

Comment 1:

My first comment is with regards to the motivation of this work. There is large motivation in understanding how to analyze admixed populations as the increased globalization should also increase the numbers of individuals who are admixed. But currently the large diverse multi-ethnic cohorts in my experience have analyzed individuals from each ancestry separately and leveraged information from each model within ancestries to increase, for example, prediction performance of polygenic risk scores¹. Typically, methods avoid the analysis of admixed populations as defining them accurately is difficult as well as a lack of methodological development of methods.

The reviewer is correct in that traditionally studies exclude data from admixed individuals or split diverse, multi-ancestry cohorts into 'homogeneous' groups for GWAS and polygenic prediction. This is primarily driven by a lack of statistical methods that can account for the complex genetic structure of diverse cohorts. However, this traditional approach is limited in that it does not fully leverage the information in diverse cohorts²⁻⁴, and relies on arbitrary groups defined largely by continental ancestry, from which the field is trying to move away^{4,5}. Furthermore, population structure is a matter of scale and is ubiquitous and can lead to biases, even in homogeneous cohorts⁶. Thus, we believe that admixture and population structure cannot and perhaps, should not, be avoided and we need to understand how these processes shape the genetic architecture of complex traits and affect its inference. Our goal in this paper is exactly that: to clarify the interpretation and potential biases in heritability estimation in admixed populations by connecting widely used estimators to the underlying data generating processes (genetic architecture and demographic history).

Much of the discussion of heritability estimation in structured populations is focused on confounding due to long-range LD and environmental stratification^{7,8}. We provide a rigorous analysis of another source of bias that has so far not received much attention (though see refs^{9,10} for this issue in non-admixed contexts). We think this will be immensely useful in the long-term for developing solutions to this problem, especially as admixed and other diverse cohorts become increasingly common.

Comment 2:

Throughout the manuscript there are a few notations that are not clearly defined when first introduced. Both m and F_{ST} are defined when they are introduced in equation 1 and the first paragraph of the results, respectively.

Thank you for catching this. We have now defined m and F_{st} in lines 79 and 129, respectively, of the revised manuscript. We have also gone through and addressed other potential notation issues.

Comment 3:

Under Equation 1, the article examines the bias of two procedure GREML and HE regression in a variety of simulations and discusses where these methods may be biased. As this article goal is to "study the behavior of $[[h^2]]_{snp}$ in admixed populations", the analysis of only two methods greatly diminishes the scope of the article for users of other software. For example, LDSC uses a method of moments estimator and methods such as GENESIS use a Bayesian spike and slab prior to estimate heritability^{2,3}. From this article, a user of these methods has a very limited understanding of how methods they might be most comfortable are impacted. Further comparisons to a broader set of methods in either the results or discussion section will provide greatly increase the scope to the average reader.

There are numerous heritability estimators and a thorough analysis of each one is beyond the scope of this paper. But we agree that the inclusion of LDSC – a widely used method that relies on summary statistics as opposed to individual-level data – would significantly improve the scope of the paper. We have now added a section on LDSC, decomposing the estimator in terms of the genic and LD components, analyzing its behavior through extensive simulations, and comparing it against GREML and HE regression. We show that LDSC is also biased in the presence of admixture structure. The direction and magnitude of this bias depends on the genetic architecture and demographic history. LDSC behaves much like GREML under the HI model (Fig. 6A) but is biased upwards under the CGF model for all traits. We believe this is due to subtle patterns of long-range LD in the first few generations of admixture under the CGF model that are not fully accounted for by adjusting for individual ancestry in the upstream GWAS (Fig. 6B). We thank the reviewer for this suggestion.

Comment 4:

The article provides a clear conclusion the biases in estimation of heritability of the SNP occur when either using HE regression or GREML under differing true genetic architecture of the admixed population. The only time when bias was near 0, was in the case of scaling the genotypes by the LD matrix, which as properly noted is almost always impossible. Therefore, for a practitioner, there is an overall lack of general conclusion to garner from this manuscript. In a previous response to reviewers, the authors expressed that they still prefer GREML over HE regression, if this is the case, then there should be more structure in the discussion section about this to provide a clearer take away message for the reader. As the current discussion, provides confusion and the message of avoiding analyses of admixed populations.

We have made several changes to the results and discussion section to reflect this point. Specifically, we suggest that users (1) account for the LD contribution when reporting the variance explained by GWAS SNPs (Line 421), (2) remove samples that might represent a very small fraction of the GWAS sample (in terms of ancestry) (Line 460), (3) report SNP heritability estimates from all three approaches to evaluate inconsistencies for sources of bias (Line 462), (4) exercise caution when using SNP heritability estimates as an upper bound of polygenic prediction accuracy, especially for traits where the genetic risk varies as a function of ancestry (Line 472), (5) can use SNP heritability estimates to define the power of GWAS discovery (Line 467).

References:

1. Ge, T., Chen, C. Y., Neale, B. M., Sabuncu, M. R. & Smoller, J. W. Phenome-wide heritability analysis of the UK Biobank. *PLoS Genet* 13, e1006711 (2017).
2. Hou, K., Bhattacharya, A., Mester, R., Burch, K. S. & Pasaniuc, B. On powerful GWAS in admixed populations. *Nature Genetics* 2021 53:12 53, 1631–1633 (2021).
3. Lin, M., Park, D. S., Zaitlen, N. A., Henn, B. M. & Gignoux, C. R. Admixed Populations Improve Power for Variant Discovery and Portability in Genome-Wide Association Studies. *Front Genet* 12, 673167 (2021).
4. Ben-Eghan, C. *et al.* Don't ignore genetic data from minority populations. *Nature* 585, 184–186 (2020).
5. National Academies of Sciences, E. and M. Using Population Descriptors in Genetics and Genomics Research: A New Framework for an Evolving Field. *Using Population Descriptors in Genetics and Genomics Research: A New Framework for an Evolving Field* 1–217 (2023) doi:10.17226/26902.
6. Zaidi, A. A. & Mathieson, I. Demographic history mediates the effect of stratification on polygenic scores. *Elife* 9, 1–30 (2020).
7. Browning, S. R. & Browning, B. L. Population structure can inflate SNP-based heritability estimates. *Am J Hum Genet* 89, 191–193 (2011).
8. Goddard, M. E., Lee, S. H., Yang, J., Wray, N. R. & Visscher, P. M. Response to browning and browning. *Am J Hum Genet* 89, 193–195 (2011).
9. Rawlik, K., Canela-Xandri, O., Olliams, J. W. & Tenesa, A. SNP heritability: What are we estimating? *bioRxiv* 2020.09.15.276121 (2020) doi:10.1101/2020.09.15.276121.

-
10. de los Campos, G., Sorensen, D. & Gianola, D. Genomic Heritability: What Is It? *PLoS Genet* 11, e1005048 (2015).

April 29, 2025

RE: GENETICS-2025-308022

Dr. Arslan A Zaidi
University of Minnesota Twin Cities
Genetics, Cell Biology, Development
6-126 MCB Building
420 Washington Ave SE
Minneapolis, Minnesota

Dear Dr. Zaidi:

Congratulations, your manuscript titled "Interpreting SNP heritability in admixed populations" is accepted for publication in GENETICS! Many thanks for submitting your research to the journal.

To Proceed to Publication:

1. Format your article according to GENETICS style: <https://academic.oup.com/genetics/pages/general-instructions>
2. Ensure that you comply with data and community resource citation guidelines: <https://academic.oup.com/genetics/pages/general-instructions#Data-Policy>
3. Upload your final files at <https://genetics.msubmit.net>
4. Add oupsupport@scipris.com and genetics.oup@novatechset.com (or the domains @scipris.com and @novatechset.com) to your email program's "safe senders" list. You will be contacted by both at various points during the production process.

Notes:

- Your currently-accepted manuscript (unedited, as submitted, reviewed, and accepted) will be published at GENETICS and deposited into PubMed as an Advance Access article. Notify sourcefiles@thegsajournals.org before signing your license if you do not wish to publish your article via Advance Access.
- We invite you to submit an original color figure related to your paper for consideration as cover art. Please email your submission to the editorial office or upload it with your final files. You can submit a small-sized image for evaluation, and if selected, the final image must be a TIFF file 2513px wide by 3263px high (8.375 by 10.875 inches; resolution of 600ppi). Please avoid graphs and small type.
- After files are sent to Oxford University Press we use SciPris to manage article licensing and payment. If you do not have a SciPris account, you will receive an email from no-reply@scipris.com to sign up to use Oxford University Press' author portal. After logging in, follow the online instructions to sign your license and arrange any payment due.

If you have any questions or encounter any problems while uploading your accepted manuscript files, please email the editorial office at sourcefiles@thegsajournals.org.

Sincerely,

Yun Li
Associate Editor
GENETICS

Approved by:
Hongyu Zhao
Senior Editor
GENETICS

Review comments (if applicable):

Reviewer #1 :

I have no further comments.

Reviewer #2 :

None